# Estimating the contribution of subclinical tuberculosis disease to transmission: An individual patient data analysis from prevalence surveys

Jon C Emery[1]*, Peter J Dodd[2], Sayera Banu[3], Beatrice Frascella[4], Frances L Garden[5,6], Katherine C Horton[1], Shahed Hossain[7], Irwin Law[8], Frank van Leth[9,10], Guy B Marks[5,11], Hoa Binh Nguyen[12], Hai Viet Nguyen[12], Ikushi Onozaki[13], Maria Imelda D Quelapio[14], Alexandra S Richards[1], Nabila Shaikh[1,15], Edine W Tiemersma[16], Richard G White[1], Khalequ Zaman[3], Frank Cobelens[17], Rein MGJ Houben[1]

[1]TB Modelling Group, TB Centre and Centre for Mathematical Modelling of Infectious Diseases, Department of Infectious Disease Epidemiology, London School of Hygiene & Tropical Medicine, London, United Kingdom; [2]School of Health and Related Research, University of Sheffield, Sheffield, United Kingdom; [3]International Centre for Diarrhoeal Disease Research, Dhaka, Bangladesh; [4]School of Public Health, Vita-Salute San Raffaele University, Milan, Italy; [5]South West Sydney Clinical Campuses, University of New South Wales, Sydney, Australia; [6]Ingham Institute of Applied Medical Research, Sydney, Australia; [7]James P. Grant School of Public Health, BRAC University, Dhaka, Bangladesh; [8]Global Tuberculosis Programme, World Health Organization, Geneva, Switzerland; [9]Department of Health Sciences, VU University, Amsterdam, Netherlands; [10]Amsterdam Public Health Research Institute, Amsterdam, Netherlands; [11]Woolcock Institute of Medical Research, Sydney, Australia; [12]National Lung Hospital, National Tuberculosis Control Program, Ha Noi, Viet Nam; [13]Research Institute of Tuberculosis, Japan Anti-Tuberculosis Association, Tokyo, Japan; [14]Tropical Disease Foundation, Makati City, Philippines; [15]Sanofi Pasteur, Reading, United Kingdom; [16]KNCV Tuberculosis Foundation, The Hague, Netherlands; [17]Department of Global Health and Amsterdam Institute for Global Health and Development, Amsterdam University Medical Centers, University of Amsterdam, Amsterdam, Netherlands

*For correspondence:
Jon.Emery@lshtm.ac.uk

## Abstract

**Background:** Individuals with bacteriologically confirmed pulmonary tuberculosis (TB) disease who do not report symptoms (subclinical TB) represent around half of all prevalent cases of TB, yet their contribution to *Mycobacterium tuberculosis* (*Mtb*) transmission is unknown, especially compared to individuals who report symptoms at the time of diagnosis (clinical TB). Relative infectiousness can be approximated by cumulative infections in household contacts, but such data are rare.

**Methods:** We reviewed the literature to identify studies where surveys of *Mtb* infection were linked to population surveys of TB disease. We collated individual-level data on representative populations for analysis and used literature on the relative durations of subclinical and clinical TB to estimate relative infectiousness through a cumulative hazard model, accounting for sputum-smear status.

Relative prevalence of subclinical and clinical disease in high-burden settings was used to estimate the contribution of subclinical TB to global *Mtb* transmission.

**Results:** We collated data on 414 index cases and 789 household contacts from three prevalence surveys (Bangladesh, the Philippines, and Viet Nam) and one case-finding trial in Viet Nam. The odds ratio for infection in a household with a clinical versus subclinical index case (irrespective of sputum smear status) was 1.2 (0.6–2.3, 95% confidence interval). Adjusting for duration of disease, we found a per-unit-time infectiousness of subclinical TB relative to clinical TB of 1.93 (0.62–6.18, 95% prediction interval [PrI]). Fourteen countries across Asia and Africa provided data on relative prevalence of subclinical and clinical TB, suggesting an estimated 68% (27–92%, 95% PrI) of global transmission is from subclinical TB.

**Conclusions:** Our results suggest that subclinical TB contributes substantially to transmission and needs to be diagnosed and treated for effective progress towards TB elimination.

**Funding:** JCE, KCH, ASR, NS, and RH have received funding from the European Research Council (ERC) under the Horizon 2020 research and innovation programme (ERC Starting Grant No. 757699) KCH is also supported by UK FCDO (Leaving no-one behind: transforming gendered pathways to health for TB). This research has been partially funded by UK aid from the UK government (to KCH); however, the views expressed do not necessarily reflect the UK government's official policies. PJD was supported by a fellowship from the UK Medical Research Council (MR/P022081/1); this UK-funded award is part of the EDCTP2 programme supported by the European Union. RGW is funded by the Wellcome Trust (218261/Z/19/Z), NIH (1R01AI147321-01), EDTCP (RIA208D-2505B), UK MRC (CCF17-7779 via SET Bloomsbury), ESRC (ES/P008011/1), BMGF (OPP1084276, OPP1135288 and INV-001754), and the WHO (2020/985800-0).

## Editor's evaluation

This important study estimates the amount of tuberculosis transmission attributable to subclinical (asymptomatic) cases at the population level. The authors rely on existing survey data on household contacts of index cases with tuberculosis, and their respective symptomatic and bacteriological status. A solid, novel approach is proposed to incorporate this existing data to produce meaningful indicators of the relative importance of subclinical tuberculosis in sustaining the global tuberculosis epidemic. In addition to being of interest to tuberculosis epidemiologists, these results will be an important resource for policymakers in the ongoing debate about tuberculosis case finding and the role of symptom screening algorithms.

## Introduction

An estimated 1.5 million people died from tuberculosis (TB) disease in 2020, and TB is on course to retake its position as the largest cause of death by a single infectious agent (*World Health Organisation, 2021*). Fuelled by ongoing transmission through exhaled or expectorated *Mycobacterium tuberculosis* (*Mtb*) bacteria, TB incidence is declining at a rate of 1–2% per annum, which is too slow given the risk and scale of mortality (*World Health Organisation, 2021*; *Ragonnet et al., 2021*), lifelong impairment (*Dodd et al., 2021*; *Alene et al., 2021*), poverty (*Pedrazzoli et al., 2021*), and macroeconomic consequences (*Silva et al., 2021*). Problematically, most *Mtb* transmission in high-incidence settings remains unaccounted for (*Dodd et al., 2021*), with less than 1-in-10 occurrences of TB explained by transmission from a known contact (*Glynn et al., 2015*).

In recent decades the prevailing paradigm in TB policy held that symptoms and infectiousness commence simultaneously as part of 'active disease' (*World Health Organization, 2020*; *World Health Organisation, 1974*; *Houben et al., 2019*). As a consequence, a policy of passive case-finding (*Golub et al., 2005*), in which individuals are expected to attend a health facility with TB-related symptoms before receiving diagnosis and treatment, was relied upon to prevent deaths from TB, which it has (*World Health Organisation, 2021*; *Mandal et al., 2017*), and reduce incidence by interrupting transmission, which it has not (*World Health Organisation, 2021*).

Over the last decade, this classic paradigm of TB has been increasingly challenged (*Barry et al., 2009*; *Drain et al., 2018*; *Behr et al., 2018*). One important advance has been the finding in population surveys that not all individuals identified with bacteriologically confirmed TB report having

symptoms such as cough at the time of screening for TB (*Onozaki et al., 2015*; *Law et al., 2020*). As such we can make a distinction between clinical and subclinical TB, where subclinical TB (sometimes referred to as 'asymptomatic' [*Frascella et al., 2021*] or 'early' TB [*Kendall et al., 2021*]) refers to individuals who have detectable *Mtb* bacteria in their sputum but do not experience, are not aware of, or do not report symptoms (*Houben et al., 2019*). In contrast, individuals with clinical TB disease report symptoms. We distinguish both disease states from *Mtb* infection, whereby individuals may test positive on a tuberculin skin test (TST) or interferon-gamma release assay (IGRA) but do not have bacteriologically confirmed disease.

Empirical data have shown that bacteriological state (i.e. whether *Mtb* is detectable in pulmonary secretions) is a strong predictor of the potential for transmission. For example, molecular epidemiological studies show that sputum smear-positive individuals (i.e. *Mtb* detected via microscopy) are 3–6 times more likely to be sources for TB disease in contacts compared to smear-negative individuals (*Behr et al., 1999*; *Hernández-Garduño et al., 2004*; *Tostmann et al., 2008*). Surveys of *Mtb* infection prevalence in household contacts provide similar values (*Grzybowski et al., 1975*). These studies focussed on passively diagnosed individuals with clinical disease. It is, however, increasingly clear that the presence of respiratory symptoms, such as a persistent cough, is not required for the exhalation of potentially *Mtb*-containing aerosols (*Patterson and Wood, 2019*; *Asadi et al., 2019*; *Leung et al., 2020*; *Dinkele et al., 2022*). Indeed, whilst recent empirical studies have suggested that tidal breathing may contribute significantly to *Mtb* transmission (*Dinkele et al., 2022*), exhalation of infectious aerosols appears unrelated to the presence of symptoms (*Theron et al., 2020*) or cough frequency (*Williams et al., 2020*) in TB patients. This supports the hypothesis that subclinical disease can contribute, potentially substantially, to transmission (*Houben et al., 2019*; *Kendall et al., 2021*; *Dowdy et al., 2013*).

A recent review found that about half of prevalent bacteriologically confirmed pulmonary TB disease is subclinical (*Frascella et al., 2021*) and it is becoming increasingly apparent that subclinical TB can persist for a long period without progressing to clinical disease (*Richards et al., 2021*; *Ku et al., 2021*). As individuals with subclinical TB will not be identified by current passive case-finding strategies, they will continue to contact susceptible individuals and, if infectious, transmit throughout their subclinical phase. It is therefore possible that those with subclinical TB may be a major contributor to ongoing, and unaccounted for, *Mtb* transmission. If this is the case, and if the ambitious goal to end TB as a global health problem by 2035 is to be met (*World Health Organisation, 2022*), TB policy needs to shift away from solely focussing on symptom-dependent case-finding (e.g. patient-initiated passive case-finding) towards strategies that are symptom-independent.

To motivate and inform such a shift in research and policy priorities, two key questions that to date remain unanswered must be addressed. Firstly, how infectious are individuals with subclinical TB compared to those with clinical TB per unit time, and, secondly, what is their contribution to overall transmission in the current TB epidemic?

In TB, data sources on the transmission potential from sputum smear-negative individuals relative to smear-positive (e.g. molecular epidemiological studies; *Behr et al., 1999*; *Hernández-Garduño et al., 2004*; *Tostmann et al., 2008*) have often been directly interpreted as relative infectiousness, which is incorrect (*Kendall, 2021*). Instead of representing the metric of interest, which is the potential for transmission per unit time for a particular group relative to a reference group (i.e. relative infectiousness), these data actually provided a relative estimate of cumulative exposure (as acknowledged by these studies' authors; *Behr et al., 1999*; *Hernández-Garduño et al., 2004*; *Tostmann et al., 2008*). Cumulative exposure is a composite of relative infectiousness per unit time and disease duration (technically duration of infectiousness), which until now have been unavailable and can be hard to disentangle from each other (*Kendall, 2021*).

In this work we look to overcome these challenges by harnessing increased understanding of the natural history and prevalence of subclinical TB and re-analysing data from existing population studies.

## Methods
### Data
To estimate the infectiousness of subclinical TB relative to clinical TB, we considered studies in which *Mtb* infection surveys were performed amongst household contacts of culture and/or nucleic acid

amplification test (NAAT) confirmed cases where data on their symptom and sputum smear status at the time of diagnosis was available. We considered only studies in which households with no index case were also surveyed for *Mtb* infection as a measure of the background rate of infection.

Such studies identified index cases using symptom-independent screening either via a TB prevalence survey (in which all individuals are screened with a chest X-ray; *World Health Organisation, 2011*) or community-wide active case-finding amongst a representative sample of a target population. Subclinical and clinical index cases were defined as being culture and/or NAAT-positive and responding negatively or positively to an initial symptom screening, respectively. Households with a single subclinical or clinical index case were defined as subclinical and clinical households, respectively. Such households were then stratified by the sputum smear status of the index case at the time of diagnosis. Background households were defined as having no index case. Finally, *Mtb* infection surveys were performed amongst all households, providing the prevalence of infection amongst each household type.

We reviewed the literature for household contact studies that measure *Mtb* infection via TST or IGRA as an outcome and provide sufficient information to stratify households by symptom and sputum smear status, including households with no index case (see Appendix 1 for the detailed search strategy). Individual, patient-level data from each of these studies were analysed to provide the prevalence of infection amongst each household type (see Appendix 1 for detailed data analysis). These data are presented in *Appendix 1—table 1*. Odds ratios (ORs) for infection in members of a household with a sputum smear-positive versus a smear-negative index case (irrespective of symptoms) and in members of a household with a clinical versus subclinical index case (irrespective of sputum smear status) were also calculated for purposes of illustration.

## Cumulative hazard model

To estimate the infectiousness of subclinical TB per unit time relative to clinical TB, we fitted a cumulative hazard model of infection to the prevalence of infection amongst each household type for each study separately using the data described above.

For each study, household contacts were pooled into five cohorts: background; subclinical and sputum smear-negative; subclinical and sputum smear-positive; clinical and sputum smear-negative; clinical and sputum smear-positive. It was assumed that each cohort is exposed to the same background hazard, reflecting the force of infection from outside the household. It was then assumed that all cohorts except the background were exposed to an additional hazard, reflecting the force of infection from the cohort's respective index cases.

The final prevalence of infection in each cohort will then depend on the background cumulative hazard $\Lambda_B$ and an additional cumulative hazard $\Lambda_I$ specific to each household type $I$ (see Appendix 1 for model equations). We use the cumulative hazard from clinical ($C$), smear-positive ($+$) index cases as a benchmark with which to define the cumulative hazards from the remaining index case types. We assume that being subclinical ($S$) or smear-negative ($-$) have separate, multiplicative effects, such that

$$\Lambda_{C-} = r_- \Lambda_{C+}, \ \Lambda_{S+} = r_s \Lambda_{C+}, \ \Lambda_{S-} = r_- r_s \Lambda_{C+},$$

where $r_s$ and $r_-$ are the subclinical and sputum smear-negative relative cumulative hazards, respectively.

### Model fitting

The model described above was fitted to the prevalence of infection in each of the five household types for each study separately. Fitting was performed in a Bayesian framework using Markov-Chain Monte Carlo methods (see Appendix 1 for further details of model fitting). We report median and 95% equal-tailed posterior intervals (PoIs).

### Relative infectiousness of subclinical TB

To infer the infectiousness of subclinical TB per unit time relative to clinical TB from our posterior estimate for the subclinical relative cumulative hazard $r_s$, we note that, assuming constant hazards, the relative cumulative hazards from index cases will depend on the product of the relative per unit time infectiousness and relative durations of infectiousness. We assume that per unit time infectiousness depends on symptom status and sputum smear status, whilst durations of infectiousness depend on symptom status only. It follows then that:

$$r_s = \alpha_s \gamma_s,$$
$$r_- = \alpha_-,$$

where $\alpha_s$ and $\alpha_-$ are the per unit time infectiousness of subclinical relative to clinical index cases and sputum smear-negative relative to smear-positive index cases, respectively, and $\gamma_s$ is the duration of infectiousness for subclinical relative to clinical index cases.

To provide a value for the duration of infectiousness of subclinical relative to clinical index cases, we used the results from a recent study that estimated the durations of subclinical and clinical TB using a Bayesian analysis of prevalence and notification data (*Ku et al., 2021*). With the result that the subclinical phase represented between 27% and 63% of the time as a prevalent case, we used a duration of subclinical TB relative to clinical TB of 0.8 (0.4–1.7, 95% PoI). We assumed that there was no difference in duration for sputum smear-negative versus smear-positive TB.

Finally, we sampled from the posterior estimate for the subclinical relative cumulative hazard and an assumed duration of disease for subclinical index cases relative to clinical index cases, providing a median and 95% equal-tailed posterior estimate for the relative infectiousness of subclinical index cases relative to clinical index cases for each study separately. Thereafter we provide a summary estimate by mixed-effects meta-analysing the individual estimates across the separate studies. Analogous results are presented for the relative infectiousness per unit time of sputum smear-negative TB relative to smear-positive TB.

## Subclinical versus clinical TB: Prevalence and bacteriological indicators

To estimate the proportion of overall transmission from subclinical TB, we first estimated the proportion of prevalent TB that is subclinical as well as the proportion of prevalent subclinical and clinical TB that is smear-positive.

We began with a recent review of TB prevalence surveys in Asia and Africa (*Frascella et al., 2021*) (see Appendix 1 for details of the search strategy). Such surveys generally performed an initial screening using both a questionnaire, which includes questions about recent symptoms typical of TB, as well a chest radiograph. Those screening positive from either method were then tested via culture and/or NAAT. A sputum smear test was often additionally performed.

We reviewed the surveys in *Frascella et al., 2021* and, for each survey where sufficient information was available, extracted the number of culture and/or NAAT confirmed cases of TB, stratified by both symptom status at initial screening and sputum smear status (see Appendix 1 for detailed data analysis). Extracted data can be found in *Appendix 1—table 2*. We defined subclinical and clinical TB as being culture and/or NAAT-positive and responding negatively or positively to an initial symptom screen, respectively, consistent with the definitions for subclinical and clinical index cases in the previous section. The most common screening question was a productive cough of greater than 2-week duration, although other diagnostic algorithms were included.

For each survey, we calculated the proportion of prevalent TB that is subclinical ($P^S_{TB}$) as well as the proportion of prevalent subclinical and clinical TB that is smear-positive ($P^+_S$ and $P^+_C$, respectively).

We performed univariate, random-effects meta-analyses on $P^S_{TB}$, $P^+_S$ and $P^+_C$. We meta-analysed the inverse logit transformed variables, before transforming the results back to proportions and presenting a central estimate and 95% prediction interval for each variable.

## The contribution of subclinical TB to transmission

To estimate the contribution of subclinical TB to transmission, we applied our estimates of relative infectiousness to the prevalence surveys that reported the required data by symptom and smear status (see Appendix 1 for further details).

All analyses were conducted using R version 4.0.3 (*R Development Core Team, 2014*). Bayesian fitting was performed in Stan version 2.21.0 (*Stan Development Team, 2021*) using RStan (*Stan Development Team, 2020*) as an interface.

No new human subject data was collected for this work, which re-analysed individual patient data collected during four observational studies in three countries. Procedures, including consent where available, are described in the original publications. Local and institutional Ethics Approval was in place for each survey, through the Department of Health (the Philippines survey; *Tupasi et al., 1999*), Institutional Review Board of the Viet Nam National Lung Hospital (Viet Nam survey; *Hoa et al., 2010*), the Ministry of Health and Family Welfare of Bangladesh as well as the Research Review Committee

and Ethics Review Committee of the iccdr,b (Bangladesh survey; *Zaman et al., 2012*), and Institutional Review Board of the Viet Nam National Lung Hospital as well as Human Research Ethics Committee of the University of Sydney (ACT3 survey; *Marks et al., 2019*). The ethics committee of the London School of Hygiene and Tropical Medicine gave ethical approval for this project (#16396).

## Sensitivity analyses

### Sensitivity analysis 1

Given the different designs of the Bangladesh (2007) prevalence survey (*Zaman et al., 2012*) (which provided only sputum smear-positive index cases) and the active case-finding trial in Viet Nam (2017) (*Marks et al., 2019*) (which provided index cases via repeated case-finding-related screening and prevalence surveys), the above analysis was repeated omitting these studies.

### Sensitivity analysis 2

As a sensitivity the above analysis was repeated with an alternative estimate for the relative duration of subclinical TB versus clinical TB using instead data from a recent systematic review and data synthesis study (*Richards et al., 2021*) and a simple competing risk model (see Appendix 1 for further details).

### Sensitivity analysis 3

To explore the impact of a differential background risk of infection amongst households with and without index cases, the analysis was repeated assuming a 50% increase in the background risk of infection for those households with an index case.

### Sensitivity analysis 4

Instead of assuming equal durations for sputum smear-positive and smear-negative TB, the above analysis was repeated assuming that sputum smear-positive TB has twice the duration of smear-negative TB.

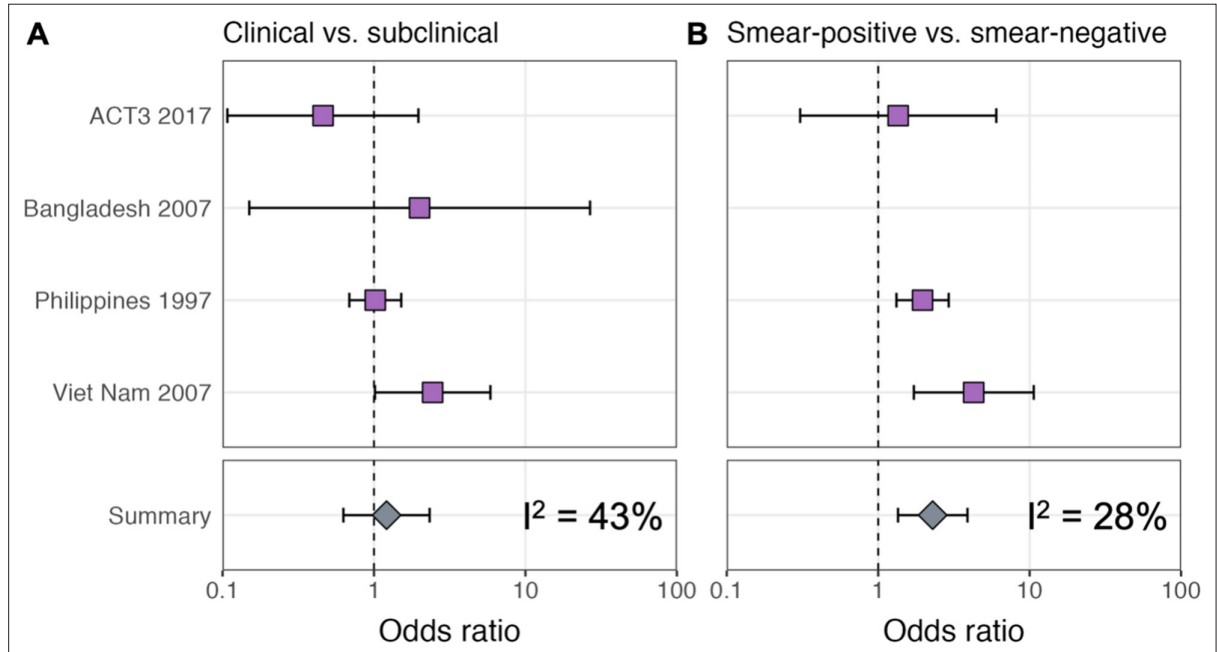

**Figure 1.** Odds ratios for infection in members of a household with a clinical versus a subclinical index case (irrespective of sputum smear-status) (**A**) and in members of a household with a sputum smear-positive versus a smear-negative index case (irrespective of symptoms) (**B**). Illustrated are central estimates and 95% confidence intervals for each study separately and the results of a mixed-effects meta-analysis. Results for sputum smear status are omitted for Bangladesh as the survey considered only sputum smear-positive individuals.

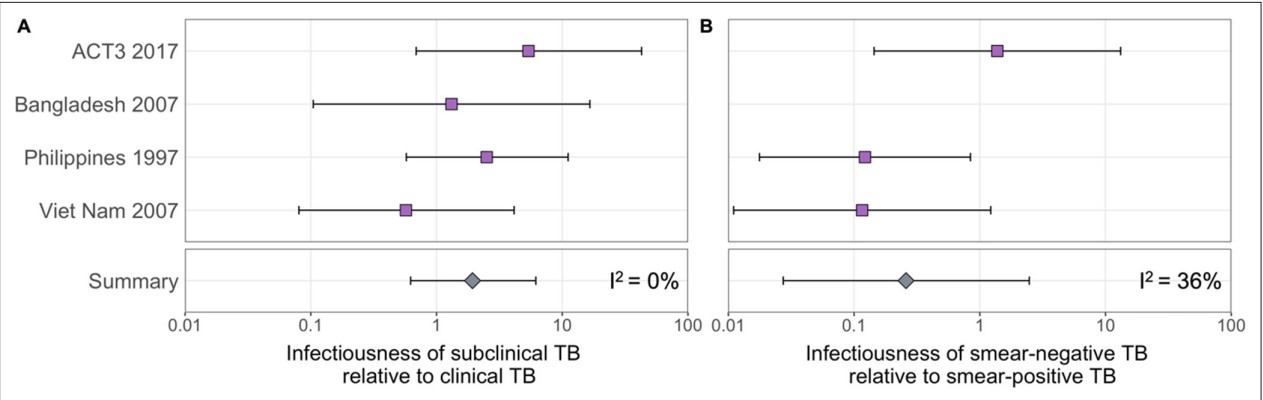

**Figure 2.** The estimated infectiousness of subclinical tuberculosis (TB) per unit time relative to clinical TB (**A**) and sputum smear-negative TB relative to smear-positive TB (**B**). Illustrated are the median and 95% confidence intervals for each study separately and the median and 95% prediction interval results from mixed-effects meta-analyses across studies with an associated measure of heterogeneity (I²).

### Sensitivity analysis 5

In the main analysis, each study was modelled separately, with the results combined using meta-analyses. As a sensitivity we model all studies simultaneously, assuming local background risks of infection for each study and global values across all studies for the remaining cumulative hazards.

## Results

### Data

Four studies were included for analysis: three prevalence surveys of TB disease with associated *Mtb* infection surveys in Viet Nam (2007) (*Hoa et al., 2010*), Bangladesh (2007) (*Zaman et al., 2012*), and the Philippines (1997) (*Tupasi et al., 1999*) and a community-wide active case-finding trial in Viet Nam (2017) (*Marks et al., 2019*).

ORs for infection in members of a household with a clinical versus subclinical index case (irrespective of sputum smear status), based on the result of their symptom screen at the time of diagnosis, are shown in *Figure 1A*. A mixed-effects meta-analysis across studies provides OR = 1.2 (0.6–2.3, 95% confidence interval [CI]). *Figure 1B* shows the OR for infection in members of a household with a sputum smear-positive versus a smear-negative index case (irrespective of symptoms), where the Bangladesh prevalence survey is omitted as this study only included smear-positive individuals. In contrast to the analysis by symptom status, evidence for a difference in cumulative infection was found by smear status, with OR = 2.3 (1.3–3.9, 95% CI), which is in line with previous estimates (*Grzybowski et al., 1975*).

### Estimating the relative infectiousness of subclinical TB

The estimated infectiousness of subclinical TB per unit time relative to clinical TB is shown in *Figure 2A*, both for each study separately as well as the mixed-effects meta-analysed result across studies of 1.93 (0.62–6.18, 95% prediction interval [PrI]). *Figure 2B* shows the analogous results for the infectiousness per unit time of sputum smear-negative versus smear-positive TB, with a summary value of 0.26 (0.03–2.47, 95% PrI). Detailed model results are shown in *Appendix 1—table 4* and *Appendix 1—figures 2–5*.

### Prevalence and bacteriological indicators for subclinical and clinical TB

Data from 15 prevalence surveys where the proportion of subclinical and clinical TB was reported by sputum smear status were included, detailed in *Appendix 1—table 2*. These represented a range of high TB burden countries in Africa (n = 5) and Asia (n = 9, with two surveys in Viet Nam). In this subset, the overall proportion of prevalent TB that is subclinical was 58% (29–82%, 95% PrI), whilst the proportion smear-positive was 33% (18–52%, 95% PrI) for subclinical TB and 53% (25–80%, 95% PrI) for clinical TB. Detailed results for each variable are shown in *Figure 3A–C*.

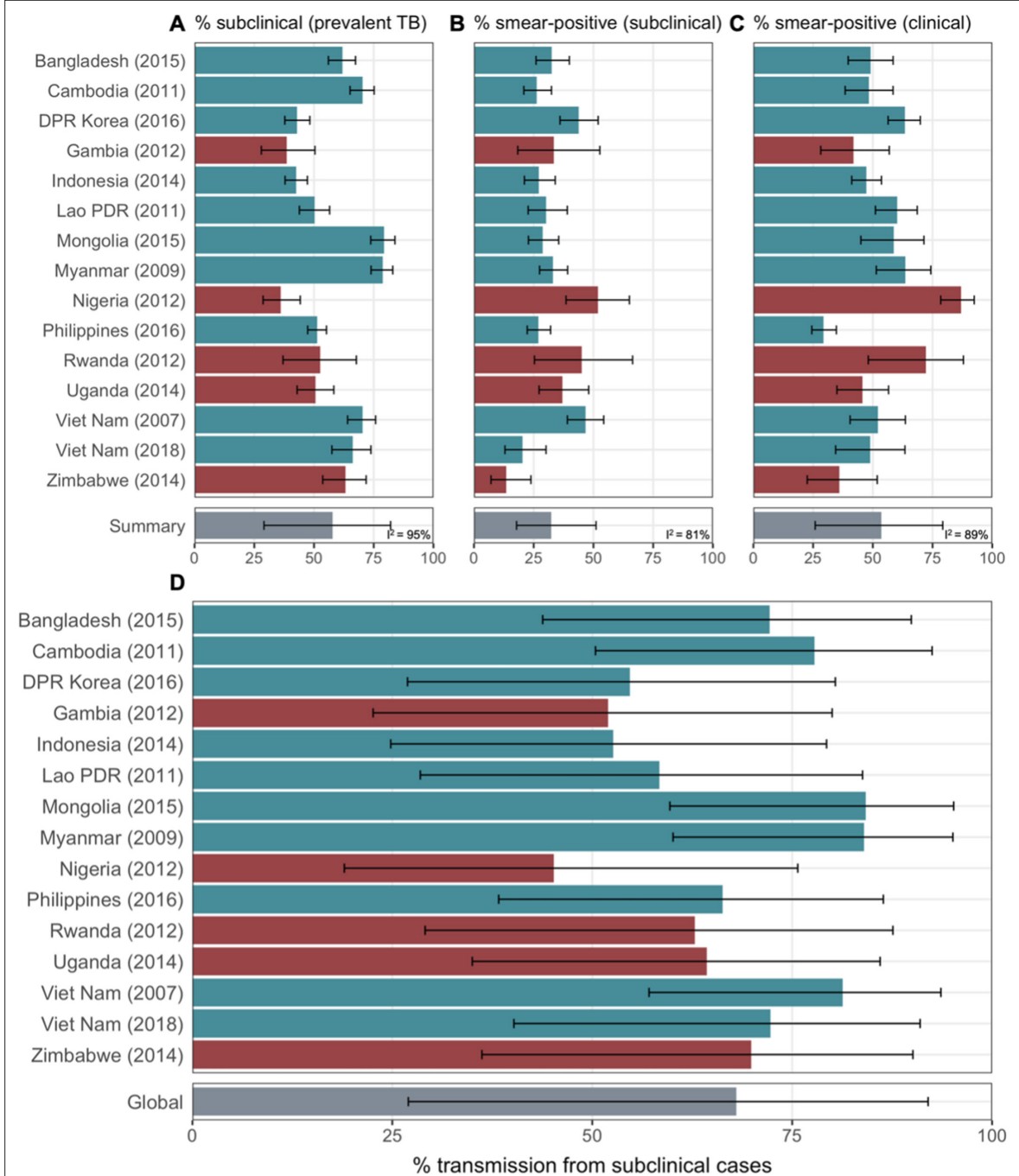

**Figure 3.** The proportion of prevalent tuberculosis (TB) that is subclinical (**A**), the proportion of subclinical TB that is smear-positive (**B**), and the proportion of clinical TB that is smear-positive (**C**) using data from prevalence surveys in Africa (red) and Asia (teal). Illustrated are median and 95% confidence intervals for each study separately and the median and 95% prediction intervals from mixed-effects meta-analyses across studies with an associated measure of heterogeneity (I²). Also shown is the estimated proportion of transmission from subclinical TB at the time of and in the location of each of the prevalence surveys in Africa and Asia (**D**). Illustrated is the median and 95% prediction intervals for each study separately as well as the global value. DPR = Democratic People's Republic; PDR = People's Democratic Republic.

## The contribution of subclinical TB to transmission: Global and country levels

We quantified the contribution of subclinical TB to ongoing *Mtb* transmission by combining the estimates for the infectiousness of subclinical TB per unit time relative to clinical TB (*Figure 2A*), the

infectiousness of sputum smear-negative TB relative to smear-positive TB (*Figure 2B*), and the proportion of prevalent TB that is subclinical and the proportion of subclinical and clinical TB that is sputum smear-positive (*Figure 3A–C*). The 14 included countries are a reasonable reflection of the geography and epidemiological characteristics of high TB burden countries in the WHO African, South-East Asia, and Western Pacific regions, which together represent around 85% of current global TB incidence (*World Health Organisation, 2021*). As such we used a summary value for the included surveys as a global estimate.

*Figure 3D* shows the results by country and globally, where 68% (27–92%, 95% PrI) of global *Mtb* transmission is estimated to come from prevalent subclinical TB, ranging from 45% (19–76%, 95% PrI) in Nigeria to 84% (60–95%, 95% PrI) in Mongolia.

## Sensitivity analyses

### Sensitivity analysis 1

The above analysis was repeated excluding two studies with methodologies that differed from the remaining two: the Bangladesh (2007) prevalence survey (*Zaman et al., 2012*) (which provided sputum smear-positive index cases only) and the active case-finding trial in Viet Nam (ACT3 [2017]) (*Marks et al., 2019*) (which provided index cases via repeated screening related to case-finding as well as prevalence surveys). Affected results are shown in *Appendix 1—figure 6*. The infectiousness of subclinical TB per unit time relative to clinical TB decreased to 1.39 (0.17–11.2, 95% PrI), and the infectiousness of sputum smear-negative TB relative to smear-positive TB decreased to 0.12 (0.03–0.53, 95% PrI), with corresponding values of 57% (10–94%, 95% PrI) of global transmission from subclinical TB, ranging from 34% (6–81%, 95% PrI) in Nigeria to 76% (28–97%, 95% PrI) in Mongolia.

### Sensitivity analysis 2

The above analysis was repeated using an alternative estimate for the relative duration of subclinical TB versus clinical TB of 0.72 (0.60–0.89, 95% PoI), from *Richards et al., 2021*. Affected results are shown in *Appendix 1—figure 7*. The infectiousness of subclinical TB per unit time relative to clinical TB increased to 2.19 (0.91–5.26, 95% PrI), with corresponding values of 71% (32–92%, 95% PrI) of global transmission from subclinical TB, ranging from 48% (25–74%, 95% PrI) in Nigeria to 86% (68–95%, 95% PrI) in Mongolia.

### Sensitivity analysis 3

The above analysis was repeated assuming that households with an index case have a 50% greater background risk of infection than households with no index case. Affected results are shown in *Appendix 1—figure 8*. The infectiousness of subclinical TB per unit time relative to clinical TB increased to 2.44 (0.60–10.06, 95% PrI) whilst the infectiousness of sputum smear-negative TB relative to smear-positive TB increased to 0.36 (0.03–4.51, 95% PrI), with corresponding values of 74% (29–95%, 95% PrI) of global transmission from subclinical TB, ranging from 53% (20–85%, 95% PrI) in Nigeria to 88% (61–97%, 95% PrI) in Mongolia.

### Sensitivity analysis 4

The above analysis was repeated assuming sputum smear-positive TB has twice the duration of smear-negative TB. Affected results are shown in *Appendix 1—figure 9*. The infectiousness of subclinical TB per unit time relative to clinical TB was largely unchanged at 1.94 (0.63–6.16, 95% PrI) whilst the infectiousness of sputum smear-negative TB relative to smear-positive TB increased to 0.51 (0.06–4.39, 95% PrI), with corresponding values of 70% (29–93%, 95% PrI) of global transmission from subclinical TB, ranging from 49% (21–79%, 95% PrI) in Nigeria to 86% (63–96%, 95% PrI) in Mongolia.

### Sensitivity analysis 5

The above analysis was repeated with all studies modelled simultaneously, assuming local background risks of infection for each study and global values across all studies for the remaining cumulative hazards. Affected results are shown in *Appendix 1—figure 10*. The infectiousness of subclinical TB per unit time relative to clinical TB decreased to 1.39 (0.50–4.02, 95% PrI) whilst the infectiousness of sputum smear-negative TB relative to smear-positive TB decreased to 0.11 (0.02–0.68, 95% PrI), with

corresponding values of 58% (20–88%, 95% PrI) of global transmission from subclinical TB, ranging from 34% (15–62%, 95% PrI) in Nigeria to 76% (52–91%, 95% PrI) in Mongolia.

## Discussion

By fitting a cumulative hazard model of infection to prevalence data amongst household contacts of subclinical and clinical index cases, we were able to provide quantitative estimates for the relative infectiousness per unit time of subclinical TB and its contribution to ongoing *Mtb* transmission. Despite wide uncertainty intervals, the raw data, as well as the results of our analysis, do not suggest subclinical TB is substantially less infectious than clinical TB. Given the high prevalence of subclinical TB found in surveys (*Frascella et al., 2021*), it is therefore likely that subclinical TB contributes substantially to ongoing *Mtb* transmission in high-burden settings.

Our results were relatively robust to the sensitivities that were performed. In two cases, that is, the removal of two studies (sensitivity analysis 1) and the use of a single model to account for all studies (sensitivity analysis 5), our estimates for the relative infectiousness of subclinical TB relative to clinical TB and the proportion of transmission from subclinical TB were lower than in the primary analysis. Our qualitative results and conclusions remain unchanged however. All other sensitivities resulted in higher estimates.

There are no other estimates for the infectiousness of subclinical TB relative to clinical TB in the literature with which to compare our results. Using data from the 2007 Viet Nam prevalence survey, however, *Nguyen et al., 2023* find that among children aged 6–10 years, those living with clinical, smear-positive TB, and those living with subclinical, smear-positive TB had similarly increased risks of TST positivity compared with those living without TB. Moreover, a recent small study from Uganda found no evidence of a difference in cumulative infection rates in household contacts of patients who did or did not report symptoms (*Baik et al., 2021*). Our results are also in keeping with recent results from whole-genome sequencing (*Xu et al., 2019*) in which 36% of individuals likely transmitted *Mtb* before symptom onset, assuming a linear SNP mutation rate. We also note that previous hypothetical modelling of subclinical TB used assumed values for relative infectiousness that are in keeping with our estimated range (*Dowdy et al., 2013*; *Arinaminpathy and Dowdy, 2015*). More broadly, recent work on SARS-CoV-2 and malaria has similarly shown how 'asymptomatic' or 'subpatent' infections can be important drivers of transmission (*Slater et al., 2019*; *Emery et al., 2020*; *Johansson et al., 2021*), meaning a role for asymptomatic transmission would not be unique to TB.

Whilst we have presented a novel approach to investigating transmission from individuals with subclinical TB using pre-existing data, limitations in our methodology remain. Identifying relative infectiousness is challenging. Our estimates rely on studies which screened a minimum of 252,000 individuals for bacteriologically confirmed TB disease and 63,000 individuals for *Mtb* infection. Even at this scale, the small number of studies and diagnosed cases of TB still leads to substantial uncertainty, highlighting the challenge faced by single studies to estimate such values (*Marks et al., 2019*; *Baik et al., 2021*). Indeed, the paucity of the data provides an estimate that is consistent with subclinical TB being more infectious than clinical TB. Whilst we consider this to be implausible, we have avoided introducing priors that rule out this possibility. Instead we would emphasise that our results reflect the uncertainty of the data. The lower bound of our estimate precludes subclinical TB being significantly less infectious than clinical TB, while there is no evidence against subclinical TB being as infectious as clinical TB. Despite such uncertainty, this study brings together the best currently available epidemiological data which, combined with appropriate analysis techniques, provides a data-driven estimate for this important question.

Although infection studies in household contacts have provided a novel window into transmission from subclinical individuals, it is not possible to establish a transmission link between presumed index cases and infections amongst household contacts using molecular methods (*Kendall, 2021*). Such household contact studies are therefore liable to biases and our study necessarily inherits such limitations. For example, whilst our model does use a background rate of infection as a baseline from which to estimate the additional force of infection from presumed index cases within the household, there remains the residual risk that certain household types may systematically contain more or less infections from transmission outside the household than on average.

An important limitation of our cumulative hazard model is the assumption that index cases only ever had the disease type they were diagnosed with during screening (e.g. sputum smear-positive,

subclinical). Instead, it is more likely that individuals will fluctuate between being, for example, subclinical and clinical (*Richards et al., 2021*). The impact such additional dynamics would have on our results remains uncertain since they would depend on the detailed model of tuberculosis natural history assumed. Such a model would require additional data to prevent the need for additional assumptions.

We estimated the contribution of subclinical TB to transmission at the population level, including transmission outside the household, using information on relative infectiousness inferred from household contact studies. A more refined estimate may need to take additional factors into account. For example, it is likely that, whilst inside the household the contact rates for subclinical and clinical individuals are likely to be similar, contact with individuals outside the household may differ (*Glynn et al., 2020*).

We defined subclinical and clinical TB as being culture and/or NAAT-positive and responding negatively or positively to an initial symptom screen, respectively. In practice subclinical and clinical TB are part of a continuous spectrum and alternative definitions could be defined according to different criteria. Here we have used the definition most closely aligned with the methodology of the majority of prevalence surveys, which is consistent with other studies of subclinical TB (*Frascella et al., 2021*) and pragmatic for inclusion of future surveys.

Meta-analyses were used to provide ranges for several quantities of interest. Whilst the heterogeneity for the relative infectiousness of subclinical and smear-negative TB were low ($I^2$ = 0% and $I^2$ = 36%, respectively), the heterogeneity for the proportion of prevalent TB that is subclinical and the proportions of subclinical and clinical TB that are smear-positive were high ($I^2$ = 95%, $I^2$ = 81% and $I^2$ = 89%, respectively). As such, we have used the more conservative prediction interval (as opposed to credible interval) to reflect this heterogeneity in the final results (*Riley et al., 2011*; *Deeks et al., 2021*).

Finally, our data are from populations with a low prevalence of HIV co-infection and the HIV status of individuals with TB was mostly unavailable, making a sub-analysis by HIV and antiretroviral (ART) status impossible. Whilst the subclinical TB presentation is likely affected by HIV in terms of duration and prevalence, it is unknown whether or by how much the relative differences in duration and prevalence between subclinical and clinical TB also change (*Frascella et al., 2021*; *National Department of Health - South Africa, 2021*). If they exist, any such differences by HIV-coinfection status are likely to be reduced by effective viral suppression, which an estimated two-thirds of people living with HIV have achieved (*UNAIDS, 2021*). So while it remains highly valuable to accumulate additional relevant data (*Baik et al., 2021*), we feel our main findings are broadly robust to this limitation.

Our observation that reported symptoms are a poor proxy for infectiousness fits with historical and contemporary observations that symptom-independent TB screening and treatment policies can reduce TB burden at higher rates than usually seen under DOTS (*Marks et al., 2019*; *Krivinka et al., 1974*). This is in keeping with increasing data showing that symptoms, in particular the classic TB symptom of cough, are not closely correlated to the amount of *Mtb* exhaled (*Williams et al., 2020*) and observations of other pathogens, including SARS-CoV-2 infection (*Emery et al., 2020*; *Johansson et al., 2021*; *Liu et al., 2020*).

Whilst earlier diagnosis (i.e. before symptom onset) will likely bring individual-level benefits in terms of mortality and extent of post-TB sequelae (*Allwood et al., 2019*), the question of potential population benefits has hampered decisions from policymakers and funders on whether to invest resources in technologies and strategies that can identify subclinical TB. Our results suggest that a non-trivial proportion of all transmission would likely be unaffected by strategies that are insensitive to subclinical TB.

As our results show that subclinical TB likely contributes substantially to transmission, an increased emphasis on symptom agnostic screening in, for example, the TB screening guidelines (*World Health Organization, 2021*) should be considered, as should the inclusion of subclinical TB in the planned update of WHO case definitions. Target Product Profiles for diagnostic tools should consider all infectious TB, regardless of whether individuals are experiencing or aware of symptoms, and interventions using such tools should be critically evaluated for their impact on *Mtb* transmission and cost-effectiveness. While symptom-independent tools exist for screening (*Williams et al., 2020*; *Marks et al., 2019*; *Yoon et al., 2017*; *Madhani et al., 2020*; *Scriba et al., 2021*; *Williams et al., 2014*), specificity, costs, and logistics remain an obstacle. In addition, individuals are usually required to produce sputum as part of a confirmatory test, which around half of eligible adults in the general

population are unable to do (*Marks et al., 2019*). Screening or diagnostic technologies that are symptom- as well as sputum-independent, while remaining low-tech and low-cost, remain the goal. Indeed, the advent of bio-aerosol measurements in TB may uncover additional infectious individuals whose sputum-based bacteriological test is negative, although these tools require validation in larger populations (*Williams et al., 2020*; *Williams et al., 2014*; *Nathavitharana et al., 2022*). Any bio-aerosol-positive, sputum-negative individuals are more likely to be subclinical and as such would mean we underestimated the contribution of subclinical TB to global *Mtb* transmission, even if their relative infectiousness may be lower than sputum-positive TB. As new diagnostic approaches are developed to capture the spectrum of TB disease, policymakers will need to decide on how to treat subclinical TB. In treatment, as in diagnosis, it is key that more tailored approaches are developed and tested so as to prevent over- or undertreatment of individuals with subclinical TB (*Esmail et al., 2022*).

## Conclusion

Subclinical TB likely contributes substantially to transmission in high-burden settings. If we are to meet EndTB targets for TB elimination, the TB community needs to develop technologies and strategies beyond passive case finding to address subclinical TB.

## Acknowledgements

The authors acknowledge the work of Dr Thelma Tupasi who led the Philippines (2007) prevalence survey and founded the Tropical Disease Foundation (Philippines), who sadly passed away in 2019.

## Additional information

### Competing interests

Peter J Dodd: has received consultancy fees from WHO (TB burden estimation) and participates as chair of SAB for NIHR grant on TBI screening. The author has no other competing interests to declare. Guy B Marks: acts as President of the International Union Against TB & Lung Disease. The author has no other competing interests to declare. Ikushi Onozaki: has received grants from National TB Program Cambodia, WHO and DFAT (Australia) and has received consulting fees from WHO Myanmar Office. The author is on the Board of the Directors, UNION IUATLD and WHO's Global Task Force on TB Impact Measurement. The author has no other competing interests to declare. Nabila Shaikh: received a grant from the Bill and Melinda Gates Foundation and owns stock/stock options in Sanofi Aventis Pharma LTD. The author has no other competing interests to declare. The other authors declare that no competing interests exist.

### Funding

| Funder | Grant reference number | Author |
|---|---|---|
| European Research Council | ERC Starting Grant No. 757699 | Jon C Emery<br>Katherine C Horton<br>Alexandra S Richards<br>Nabila Shaikh<br>Richard G White |
| Foreign, Commonwealth and Development Office | | Katherine C Horton |
| UK Government | | Katherine C Horton |
| UK Medical Research Council | MR/P022081/1 | Peter J Dodd |
| Wellcome Trust | 218261/Z/19/Z | Richard G White |
| National Institutes of Health | 1R01AI147321-01 | Richard G White |

| Funder | Grant reference number | Author |
|---|---|---|
| European and Developing Countries Clinical Trials Partnership | RIA208D-2505B | Richard G White |
| UK Medical Research Council | CCF17-7779 | Richard G White |
| Economic and Social Research Council | ES/P008011/1 | Richard G White |
| Bill & Melinda Gates Foundation | OPP1084276 | Richard G White |
| Bill & Melinda Gates Foundation | OPP1135288 | Richard G White |
| Bill & Melinda Gates Foundation | INV-001754 | Richard G White |
| World Health Organization | 2020/985800-0 | Richard G White |

The funders had no role in study design, data collection and interpretation, or the decision to submit the work for publication. For the purpose of Open Access, the authors have applied a CC BY public copyright license to any Author Accepted Manuscript version arising from this submission.

## Author contributions

Jon C Emery, Software, Formal analysis, Investigation, Visualization, Methodology, Writing – original draft, Writing – review and editing; Peter J Dodd, Software, Formal analysis, Methodology, Writing – review and editing; Sayera Banu, Katherine C Horton, Shahed Hossain, Irwin Law, Frank van Leth, Guy B Marks, Hoa Binh Nguyen, Ikushi Onozaki, Maria Imelda D Quelapio, Alexandra S Richards, Edine W Tiemersma, Richard G White, Khalequ Zaman, Writing – review and editing; Beatrice Frascella, Frances L Garden, Hai Viet Nguyen, Formal analysis, Writing – review and editing; Nabila Shaikh, Visualization, Writing – review and editing; Frank Cobelens, Methodology, Writing – review and editing; Rein MGJ Houben, Conceptualization, Formal analysis, Methodology, Writing – review and editing

## Author ORCIDs

Jon C Emery http://orcid.org/0000-0001-6644-7604
Rein MGJ Houben https://orcid.org/0000-0003-4132-7467

## Ethics

No new human subject data was collected for this work, which re-analysed individual patient data collected during four observational studies in three countries. Procedures, including consent where available, are described in the original publications. Local and institutional Ethics Approval was in place for each survey, through the Department of Health (Philippines survey [Tupasi et al., 1999]), Institutional Review Board of the Viet Nam National Lung Hospital (Viet Nam survey [Hoa et al., 2010]), the Ministry of Health and Family Welfare of Bangladesh as well as the Research Review Committee and Ethics Review Committee of the iccdr,b (Bangladesh survey [Zaman et al., 2012]) and Institutional Review Board of the Viet Nam National Lung Hospital as well as Human Research Ethics Committee of the University of Sydney (ACT3 study [Marks et al., 2019]). The Ethics committee of the London School of Hygiene and Tropical Medicine gave ethical approval for this project (#16396).

## Decision letter and Author response

Decision letter https://doi.org/10.7554/eLife.82469.sa1
Author response https://doi.org/10.7554/eLife.82469.sa2

# Additional files

## Supplementary files
• MDAR checklist

## Data availability

Replication data and analysis scripts are available at GitHub (copy archived at *Emery JC, 2022*).

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

## Appendix 1

### Supplementary methods

Search strategy

We sought studies in which *Mtb* infection surveys were performed amongst household contacts of bacteriologically confirmed index cases, where data on their symptom and sputum smear status at the time of diagnosis was available.

We began with a recent systematic review of population-based TB prevalence surveys completed since 1990, with reports or articles publicly available through August 2019 (*Frascella et al., 2021*). Surveys were included if both a symptom screening interview and X-ray were performed on all eligible participants and if surveys reported the proportion of bacteriologically confirmed cases by screening modality as well as the proportion of bacteriologically confirmed cases that were negative on symptom screening (see *Frascella et al., 2021* for full details of the review process). We then reviewed the reports of the 28 national and subnational TB prevalence surveys included for quantitative analysis in *Frascella et al., 2021* and identified three such studies that were conducted alongside *Mtb* infection surveys amongst household contacts: Viet Nam (2007) (*Hoa et al., 2010*), Bangladesh (2007) (*Zaman et al., 2012*), and the Philippines (1997) (*Tupasi et al., 1999*). Authors of these studies and affiliated institutions were then invited to collaborate using original, individual-level data and all accepted.

In addition to prevalence surveys, we also considered active case-finding studies with associated household infection surveys with which to measure any resultant impact on transmission. A non-systematic review of the literature identified one such study in Viet Nam: ACT3 (2017) (*Marks et al., 2019*). Again the authors of this study and affiliated institutions were invited to collaborate with original, individual-level data and accepted.

Data analysis

Index cases were identified in the three prevalence surveys (Viet Nam [2007] [*Hoa et al., 2010*], Bangladesh [2007] [*Zaman et al., 2012*], and the Philippines [1997] [*Tupasi et al., 1999*]) via culture and/or NAAT and defined as subclinical or clinical depending on whether they responded negatively or positively to an initial symptom screening, respectively. Index cases were further stratified by their sputum smear status at the time of diagnosis.

Households with multiple co-prevalent index cases were excluded from the analysis to retain the premise of the analysis which links infections above the community level to the characteristics of the single index case. Co-prevalent cases were absent in one study (ACT3; *Marks et al., 2019*) and below 10% in other studies, limiting the impact on power or introduction of bias.

Linked records were then used to stratify participants of the associated *Mtb* infection survey into different household types depending on the status of the index case: background (no index case); subclinical and sputum smear-negative; subclinical and sputum smear-positive; clinical and sputum smear-negative; clinical and sputum smear-positive. For each household type the total number of contacts and number of TST or IGRA-positive contacts was extracted, shown in *Appendix 1—table 1*.

Index cases were identified in ACT3 (*Marks et al., 2019*) either through routine passive case-finding, in any of the TB screening rounds as part of active case-finding, or in the TB prevalence surveys used to measure the impact of such screening. Those identified through passive case-finding were designated clinical, whilst those identified either through screening or the prevalence surveys were stratified as subclinical or clinical depending on whether they responded negatively or positively to an initial symptom screening, respectively. Index cases were further stratified by their sputum smear status at the time of diagnosis.

The same approach to that described above was then used to find the total number of household contacts and the number of TST or IGRA-positive contacts for each household type, also shown in *Appendix 1—table 1*.

**Appendix 1—table 1.** Summary of the relevant data from studies in which *Mtb* infection surveys were performed amongst household contacts of culture and/or nucleic acid amplification ttest

(NAAT) confirmed cases where information on their symptom and sputum smear status at the time of diagnosis was available.

A negative/positive response to 'symptoms' defines subclinical/clinical tuberculosis (TB) in the corresponding study. Infected = number of tuberculin skin test (TST) or interferon-gamma release assay (IGRA)-positive household contacts; Contacts = number of household contacts with a TST or IGRA result; NA = not applicable

| | Subclinical | | | | Clinical | | | | |
| | Background | | Smear-negative | | Smear-positive | | Smear-negative | | Smear-positive | | |
| Study | Infected | Contacts | Infected | Contacts | Infected | Contacts | Infected | Contacts | Infected | Contacts | Symptoms |
|---|---|---|---|---|---|---|---|---|---|---|---|
| ACT3 2017 (*Marks et al., 2019*) | 128 | 2893 | 2 | 8 | 2 | 10 | 1 | 16 | 4 | 27 | Cough >2 wk |
| Bangladesh 2007 (*Zaman et al., 2012*) | 702 | 17,566 | NA | NA | 1 | 5 | NA | NA | 3 | 9 | Any cough |
| Philippines 1997 (*Tupasi et al., 1999*) | 3823 | 20,259 | 48 | 227 | 32 | 82 | 23 | 108 | 34 | 109 | Cough >2 wk |
| Vietnam 2007 (*Hoa et al., 2010*) | 1556 | 21,298 | 3 | 59 | 5 | 28 | 4 | 42 | 16 | 59 | Cough >2 wk |

## Cumulative hazard model

### Model equations

The prevalence of infection in background households (i.e. with no index case) is given by

$$P_B = 1 - e^{-\Lambda_B}$$

where $\Lambda_B$ is the cumulative hazard from the background, representing transmission outside the household. The prevalence of infection in households with an index case is given by

$$P_I = 1 - e^{-\Lambda_B} e^{-\Lambda_I}$$

where $\Lambda_I$ is the cumulative hazard from index case type $I$ = *subclinical and smear-negative (S-); subclinical and smear-positive (S+); clinical and smear-negative (C-); clinical and smear-positive (C).*

The cumulative hazard from clinical and smear-positive index cases is used as a benchmark to define the cumulative hazards from the remaining index case types. We assume that being subclinical or smear-negative have separate, multiplicative effects, such that

$$\Lambda_{C-} = r_- \Lambda_{C+}, \ \Lambda_{S+} = r_s \Lambda_{C+}, \ \Lambda_{S-} = r_- r_s \Lambda_{C+},$$

where $r_s$ and $r_-$ are the subclinical and sputum smear-negative relative cumulative hazards, respectively.

### Model fitting

The model was fitted to the prevalence of infection in each of the five household types for each study separately. Fitting was performed in a Bayesian framework using Markov-Chain Monte Carlo methods. We use binomial distributions for the prevalence in the likelihood and estimate the following parameters: the background cumulative hazard ($\Lambda_B$); the cumulative hazard from clinical, smear-positive index cases ($\Lambda_{C+}$), the subclinical relative cumulative hazard ($r_s$), and the sputum smear-negative relative cumulative hazard ($r_-$). We use truncated gamma and normal distributions as weak priors:

$$\Lambda_B \sim gamma(\text{alpha} = 2, \text{beta} = 20),$$
$$\Lambda_{C+} \sim normal(\text{alpha} = 2, \text{beta} = 20),$$
$$r_s \sim normal(\text{mu} = 1, \text{sigma} = 20),$$
$$r_- \sim normal(\text{mu} = 0.2, \text{sigma} = 20)$$

A total of 50,000 iterations were performed for each study, the first 25,000 of which were discarded as burn-in. Model fit, trace, correlation, and autocorrelation plots were used to ensure model suitability and convergence. We report median and 95% equal-tailed posterior intervals (PoIs).

## Subclinical versus clinical TB: Prevalence and bacteriological indicators

### Search strategy

We sought TB prevalence surveys where data on the symptom and sputum smear status at the time of diagnosis was available for those identified in the survey. We began again with the recent systematic review of population-based TB prevalence surveys (*Frascella et al., 2021*), all of which included information on the symptom status at the time of diagnosis of those identified in the survey. We again reviewed the reports of the 28 national and subnational TB prevalence surveys included for quantitative analysis in *Frascella et al., 2021* and identified 14 such studies that also included information on the sputum smear status at the time of diagnosis of those identified in the survey. Data from the second TB prevalence survey in Viet Nam in 2018 (*Nguyen et al., 2020*), which was not included in *Frascella et al., 2021*, were additionally included.

### Data analysis

From the respective survey reports, we extracted the symptom threshold used for initial symptom screening, the total number of individuals screened and the number of identified cases that were subclinical and sputum smear-negative; subclinical and sputum smear-positive; clinical and sputum smear-negative; clinical and sputum smear-positive. Results of the data extraction are shown in *Appendix 1—table 2*.

**Appendix 1—table 2.** Data extracted from 15 prevalence where sufficient information on sputum smear status at the time of diagnosis was available.

The 'symptom threshold' used for initial symptom screening is the metric used here to define subclinical (negative) and clinical (positive). Neg = negative, Pos = positive.

| Survey setting (ref) | Year | Subclinical smear neg. | Subclinical smear pos. | Clinical smear neg. | Clinical smear pos. | Number screened | Symptom threshold |
|---|---|---|---|---|---|---|---|
| Viet Nam (*Nguyen et al., 2020*) | 2018 | 67 | 17 | 22 | 21 | 61,763 | Cough >2 wk |
| Viet Nam (*Ministry of Health - Vietnam, 2008*) | 2007 | 87 | 76 | 33 | 36 | 94,179 | Productive cough >2 wk |
| Myanmar (*Ministry of Health - Myanmar, 2010*) | 2009 | 164 | 81 | 24 | 42 | 51,367 | Any symptom |
| Lao PDR (*Law et al., 2015*) | 2011 | 83 | 36 | 47 | 71 | 39,212 | Cough >2 wk and/or other |
| Cambodia (*Ministry of Health - Cambodia, 2011*) | 2011 | 163 | 58 | 48 | 45 | 37,417 | Cough >2 wk and/or other |
| Gambia (*Ministry of Health and Social Welfare - The Gambia, 2011*) | 2012 | 18 | 9 | 25 | 18 | 43,100 | Cough >2 wk and/or other |
| Rwanda (*Ministry of Health - Rwanda, 2014*) | 2012 | 11 | 9 | 5 | 13 | 43,128 | Any symptom |
| Nigeria (*Federal Republic of Nigeria, 2012*) | 2012 | 25 | 27 | 12 | 80 | 44,186 | Cough >2 wk |
| Indonesia (*Ministry of Health, Republic of Indonesia, 2015*) | 2014 | 132 | 49 | 129 | 116 | 67,944 | Cough >2 wk and/or other |
| Uganda (*The Republic of Uganda, 2015*) | 2014 | 51 | 30 | 43 | 36 | 41,154 | Cough >2 wk |
| Zimbabwe (*Ministry of Health and Child Care – Zimbabwe, 2014*) | 2014 | 58 | 9 | 25 | 14 | 33,736 | Any symptom |
| Bangladesh (*DGHS Ministry of Health and Family Welfare - Bangladesh, 2015*) | 2015 | 116 | 56 | 54 | 52 | 98,710 | Cough >2 wk and/or other |
| Mongolia (*Ministry of Health - Mongolia, 2016*) | 2015 | 139 | 56 | 21 | 30 | 50,309 | Cough >2 wk |
| DPR Korea (*Democratic People's Republic of Korea, 2016*) | 2016 | 82 | 64 | 71 | 123 | 60,683 | Cough >2 wk and/or other |
| Philippines (*Department of Health - Philippines, 2016*) | 2016 | 231 | 85 | 212 | 88 | 46,689 | Cough >2 wk and/or other |

## The contribution of subclinical TB to transmission

We combined our estimates for the relative infectiousness of subclinical TB per unit time relative to clinical TB ($\alpha_S$), the relative infectiousness of sputum smear-negative TB relative to smear-positive TB ($\alpha_-$), the meta-analysed proportion of prevalent TB that is subclinical ($P^S_{TB}$), and the proportion of prevalent subclinical and clinical TB that is smear-positive ($P^+_S$ and $P^+_C$, respectively) to estimate the per unit time contribution of subclinical TB to overall transmission ($P^S_{Tx}$):

$$P^S_{Tx} = \frac{(P^+_S \alpha_s + (1 - P^+_S)\alpha_s \alpha_-)P^S_{TB}}{(P^+_S \alpha_s + (1 - P^+_S)\alpha_s \alpha_-)P^S_{TB} + (P^+_C + (1 - P^+_C)\alpha_-)(1 - P^S_{TB})}$$

To this end, we used the posterior distributions for $\alpha_S$ and $\alpha_-$ from the earlier model fitting. We also modelled $P^+_S$, $P^+_C$ and $P^S_{TB}$ as normal distributions with means and variances taken from the univariate meta-analysis described above. The expression for the contribution of subclinical TB to overall transmission was then evaluated using $10^7$ samples where we report the median and equal-tailed 95% prediction intervals.

The above was then re-performed on a survey-by-survey basis. Here, $P^+_S$, $P^+_C$ and $P^S_{TB}$ were modelled separately for each survey and assumed to be distributed binomially. The distributions used for $\alpha_S$ and $\alpha_-$ remained unchanged.

## Sensitivity analyses

### Sensitivity analysis 2

We calculate the duration of infectiousness for subclinical relative to clinical index cases using the model and transition values from *Richards et al., 2021*. The model is shown in *Appendix 1—figure 1A* with associated transition values, which are also detailed in *Appendix 1—table 3*. Disease durations are given by the inverse sum of all transitions out of subclinical (regression or progression) or clinical disease (regression, diagnosis, and treatment or death). We find durations of 5.4 mo (4.6–6.7 mo, 95% PoI) and 7.5 mo (7.0–8.2 mo, 95% PoI) for subclinical and clinical TB, respectively (*Appendix 1—figure 1B*), giving a relative duration of subclinical versus clinical TB of 0.72 (0.60–0.89, 95% PoI).

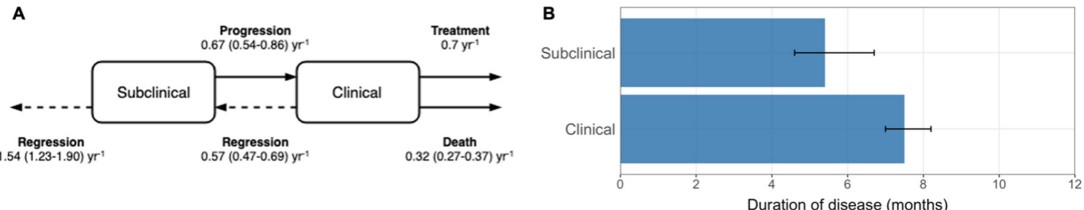

**Appendix 1—figure 1.** Competing risk model (**A**) with transition rates from *Richards et al., 2021* used to estimate the durations of subclinical and clinical tuberculosis (TB) (**B**).

**Appendix 1—table 3.** Progression and regression parameter values taken from (*Richards et al., 2021*) used to estimate the durations of subclinical and clinical tuberculosis (TB) using the competing risk method detailed in the main text.

See *Richards et al., 2021* for data sources and methods for estimating the above parameters.

| Parameter | Value (95% posterior interval) | Units |
| --- | --- | --- |
| Regression from subclinical | 1.54 (1.23–1.90) | Per year |
| Progression from subclinical | 0.67 (0.54–0.86) | Per year |
| Regression from clinical | 0.57 (0.47–0.69) | Per year |
| Treatment from clinical | 0.70 | Per year |
| Death from clinical | 0.32 (0.27–0.37) | Per year |

## Supplementary results
## Estimating the relative infectiousness of subclinical TB
Detailed model results

**Appendix 1—table 4.** Posterior summary statistics for each model.

Shown are the effective sample size (n_eff); the 'R hat' statistic (Rhat); sample mean (mean); Monte Carlo standard error (mcse); sample standard deviation (sd); and sample quantiles (2.5%, 50%, 97.5%).

| | n_eff | Rhat | Mean | mcse | sd | 2.5% | 50% | 97.5% |
|---|---|---|---|---|---|---|---|---|
| *Viet Nam* | | | | | | | | |
| lambda_B | 16,683 | 1 | 0.076 | 0.000 | 0.002 | 0.072 | 0.076 | 0.080 |
| lambda_Cp | 14,000 | 1 | 0.222 | 0.001 | 0.078 | 0.089 | 0.216 | 0.393 |
| r_s | 11,465 | 1 | 0.653 | 0.006 | 0.610 | 0.050 | 0.524 | 1.988 |
| r_n | 15,677 | 1 | 0.195 | 0.002 | 0.197 | 0.006 | 0.140 | 0.682 |
| *Philippines* | | | | | | | | |
| lambda_B | 12,836 | 1 | 0.209 | 0.000 | 0.003 | 0.203 | 0.209 | 0.216 |
| lambda_Cp | 8107 | 1 | 0.145 | 0.001 | 0.065 | 0.028 | 0.142 | 0.281 |
| r_s | 4532 | 1 | 2.644 | 0.044 | 2.940 | 0.683 | 1.910 | 9.700 |
| r_n | 15,022 | 1 | 0.172 | 0.001 | 0.131 | 0.010 | 0.145 | 0.484 |
| *ACT3* | | | | | | | | |
| lambda_B | 13,924 | 1 | 0.046 | 0.000 | 0.004 | 0.038 | 0.046 | 0.054 |
| lambda_Cp | 10,476 | 1 | 0.058 | 0.001 | 0.055 | 0.002 | 0.042 | 0.203 |
| r_s | 9133 | 1 | 6.843 | 0.074 | 7.052 | 0.612 | 4.406 | 27.192 |
| r_n | 7628 | 1 | 2.696 | 0.049 | 4.273 | 0.143 | 1.337 | 15.314 |
| *Bangladesh* | | | | | | | | |
| lambda_B | 10,804 | 1 | 0.041 | 0.000 | 0.002 | 0.038 | 0.041 | 0.044 |
| lambda_Cp | 11,415 | 1 | 0.349 | 0.002 | 0.240 | 0.037 | 0.297 | 0.944 |
| r_s | 7640 | 1 | 2.101 | 0.036 | 3.174 | 0.076 | 1.113 | 10.723 |

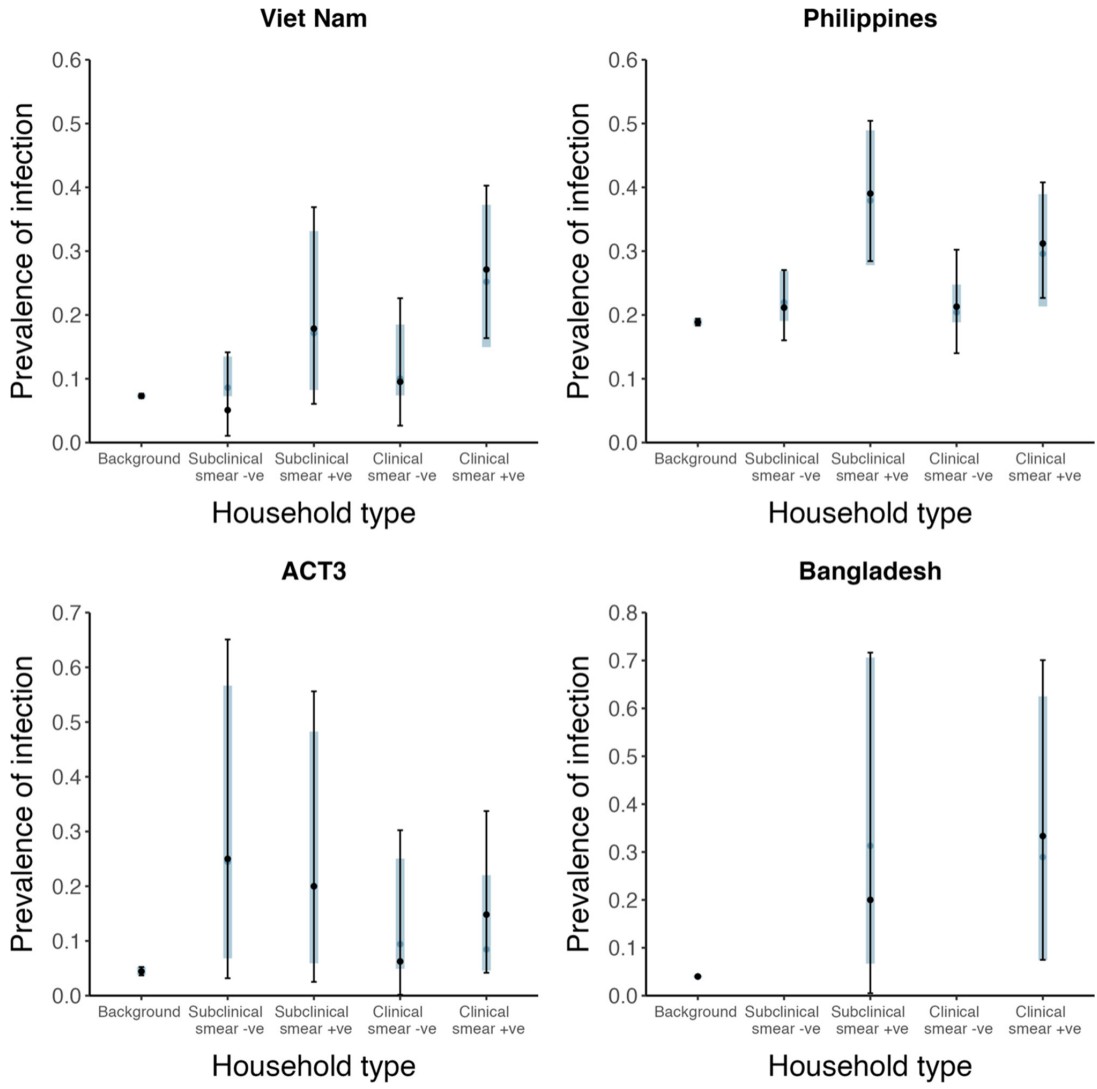

**Appendix 1—figure 2.** Model fits for each model. Shown are prevalence of infection in members of households with different index case types (background, subclinical and smear-negative, subclinical and smear-positive, clinical and smear-negative, clinical and smear-positive). Error bars show median and 95% credible intervals. Shaded regions show posterior median and 95% posterior intervals. +ve = positive, -ve = negative.

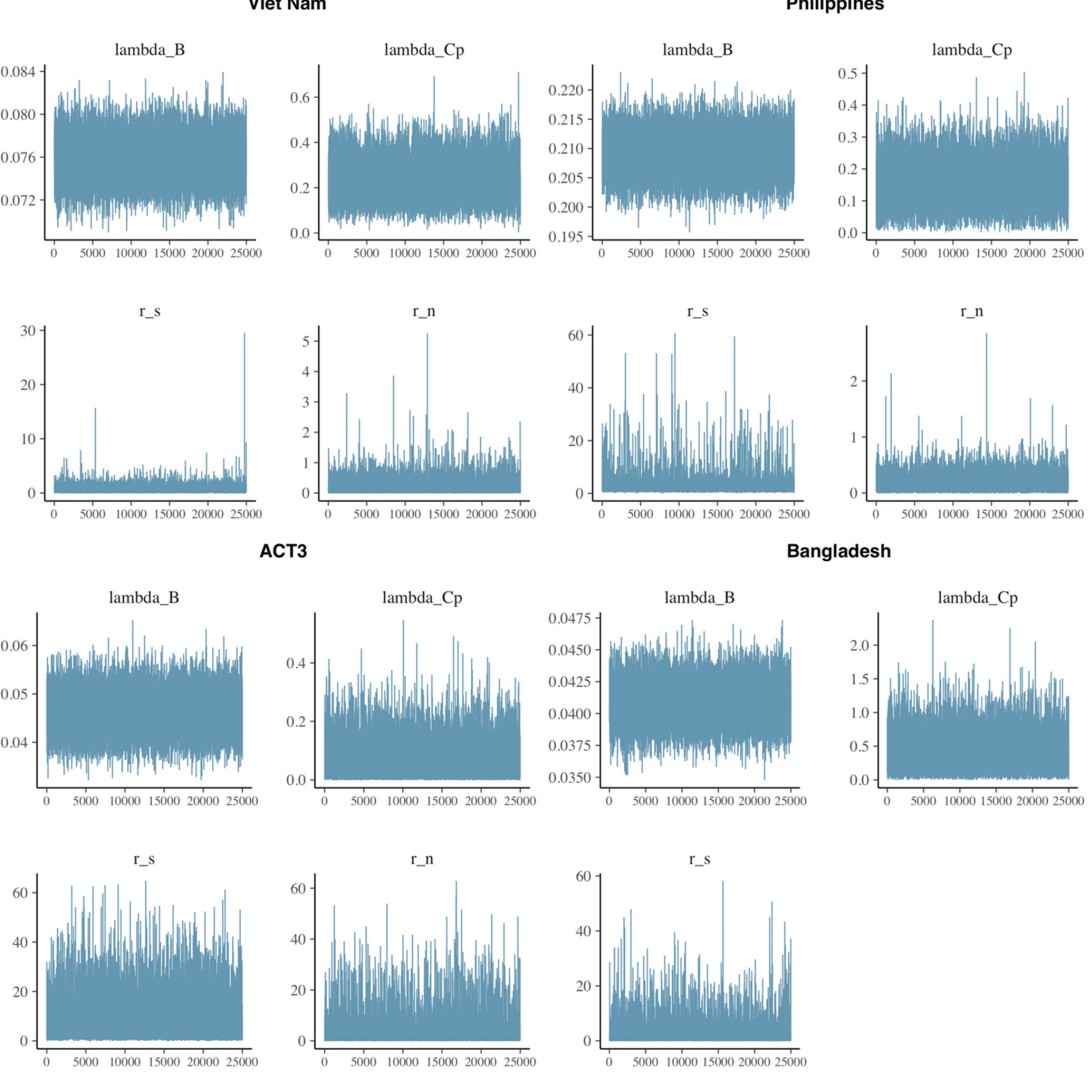

**Appendix 1—figure 3.** Trace plots for each model.

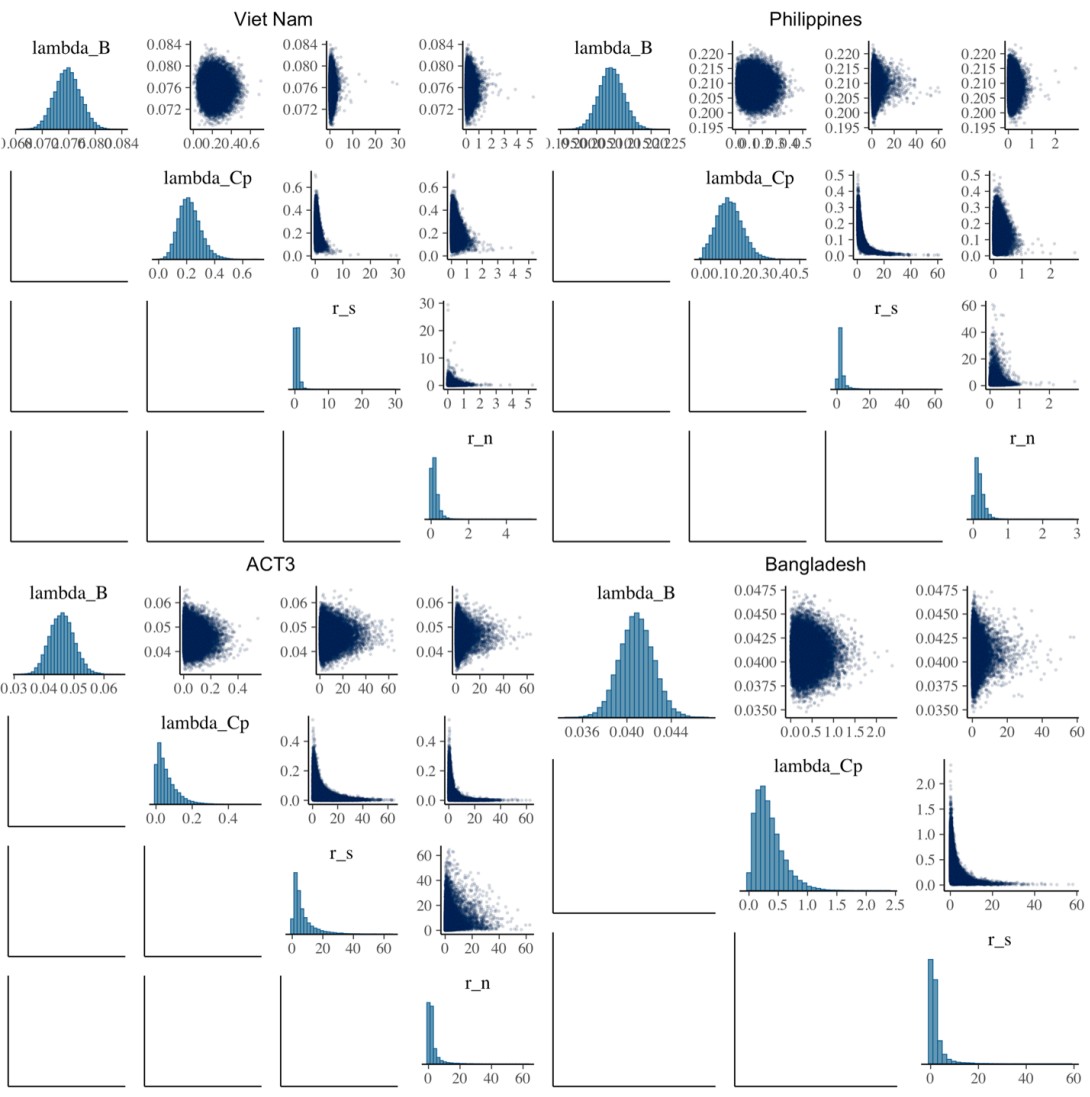

**Appendix 1—figure 4.** Correlation plots for each model.

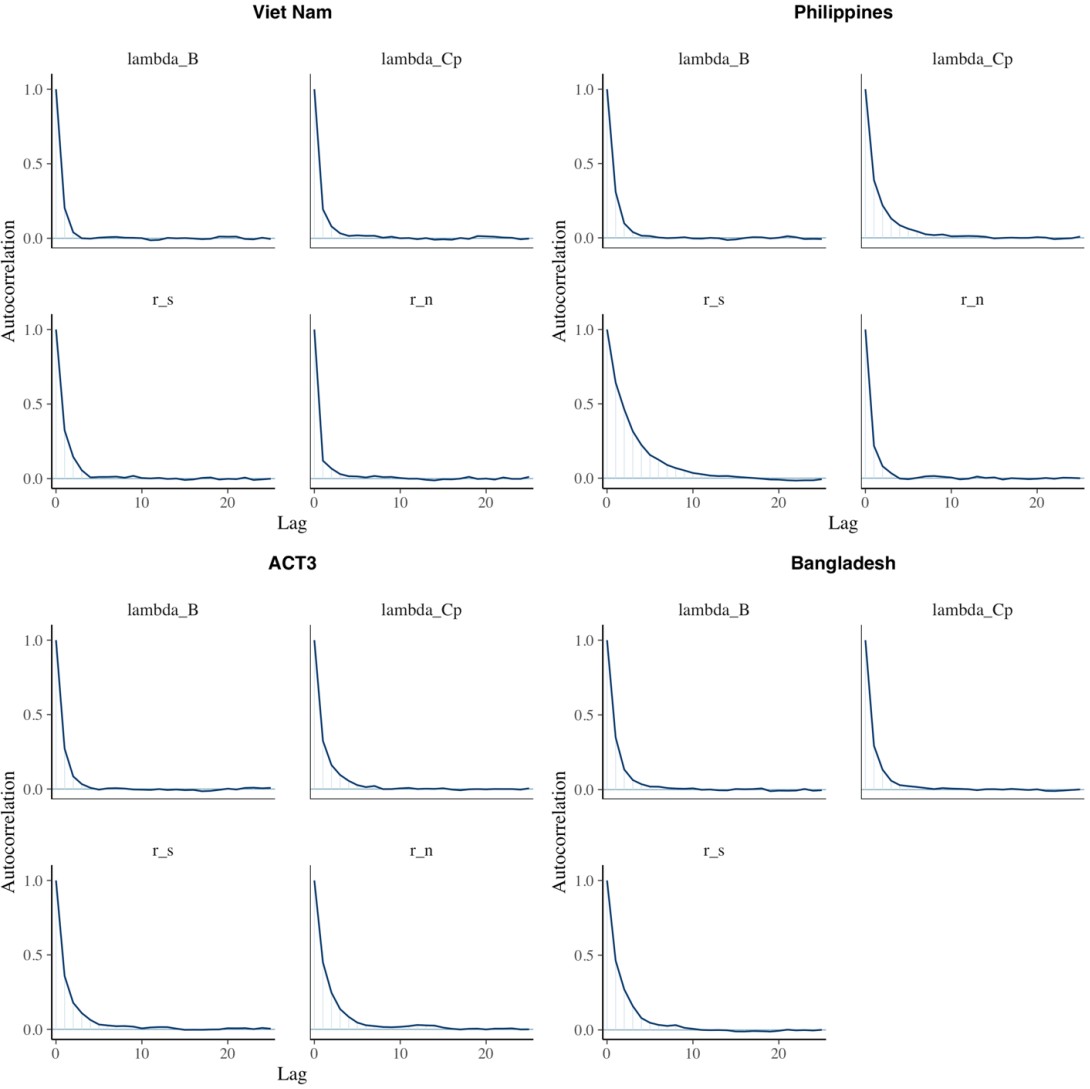

**Appendix 1—figure 5.** Autocorrelation plots for each model.

## Sensitivity analyses

### Sensitivity analysis 1

Omitting Bangladesh (2007) (*Zaman et al., 2012*) and ACT3 (2017) (*Marks et al., 2019*) from the analysis.

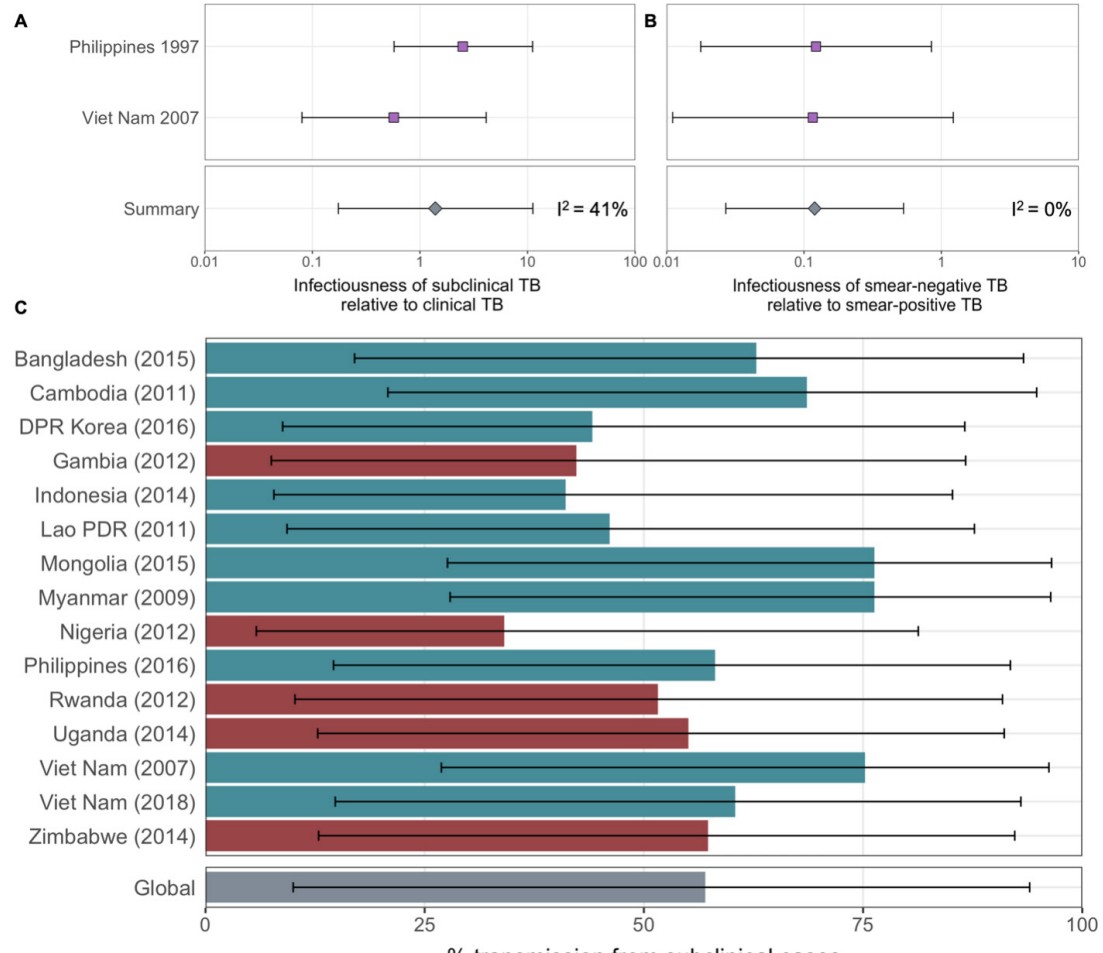

**Appendix 1—figure 6.** Affected results for sensitivity analysis 1. Figure details for A, B and C are as per *Figures 2A, B and 3D* in the main text, respectively.

## Sensitivity analysis 2

Using an alternative duration of subclinical TB relative to clinical TB from *Richards et al., 2021*.

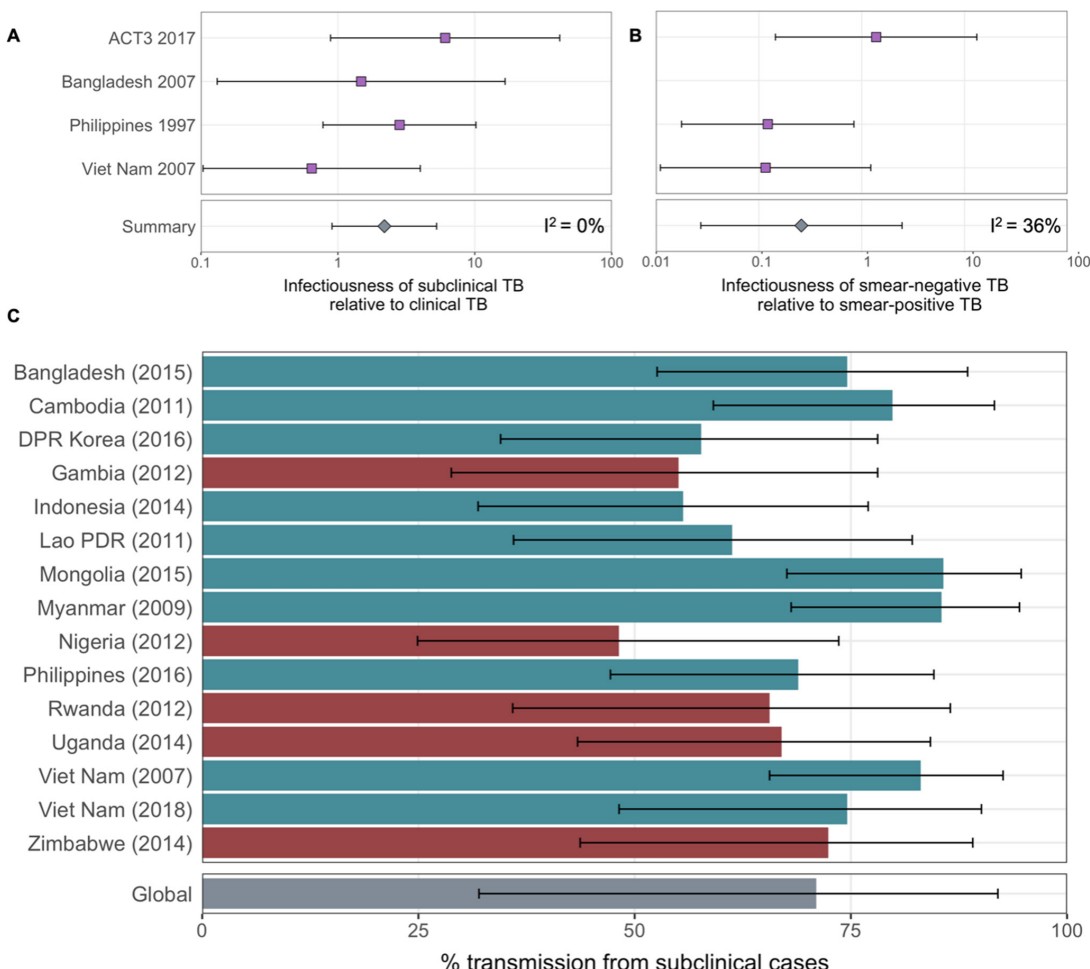

**Appendix 1—figure 7.** Affected results for sensitivity analysis 2. Figure details for A, B and C are as per *Figures 2A, B and 3D* in the main text, respectively.

## Sensitivity analysis 3

Assuming infected households have a 50% greater background risk of infection than non-infected households.

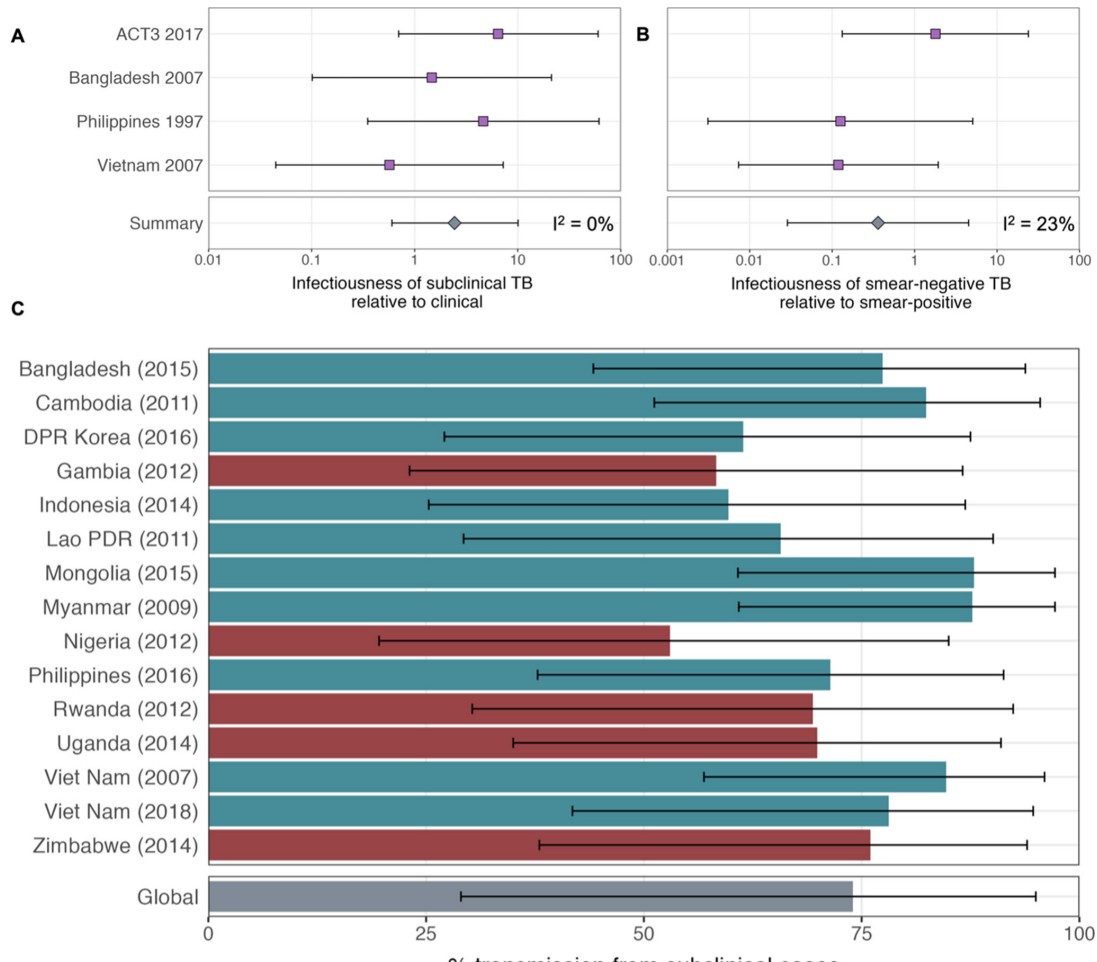

**Appendix 1—figure 8.** Affected results for sensitivity analysis 3. Figure details for A, B and C are as per *Figures 2A, B and 3D* in the main text, respectively.

## Sensitivity analysis 4
Assuming sputum smear-positive TB has twice the duration of smear-negative TB.

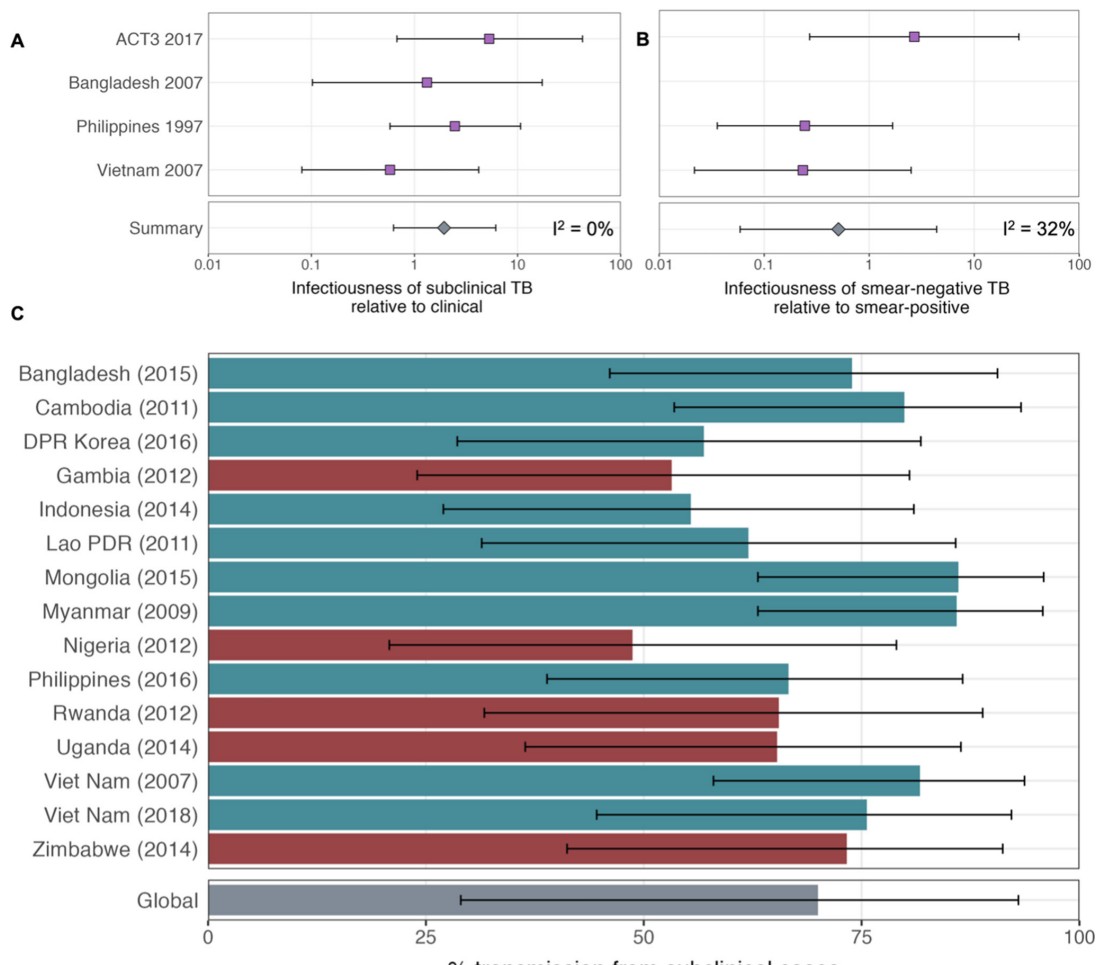

**Appendix 1—figure 9.** Affected results for sensitivity analysis 4. Figure details for A, B and C are as per *Figures 2A, B and 3D* in the main text, respectively.

## Sensitivity analysis 5

Modelling all studies simultaneously, assuming local background risks of infection for each study and global values across all studies for the remaining cumulative hazards.

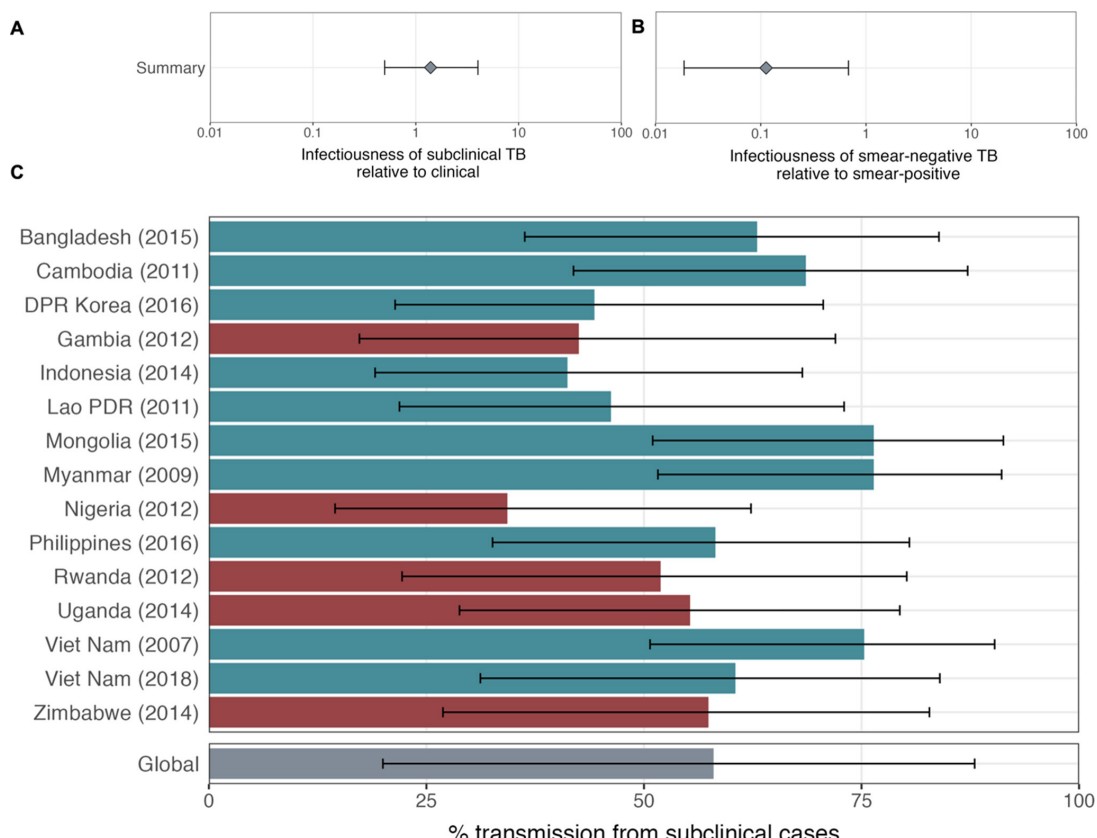

**Appendix 1—figure 10.** Affected results for sensitivity analysis 5. Figure details for A, B and C are as per *Figures 2A, B and 3D* in the main text, respectively.

