## [Editor Report]

This important study estimates the amount of tuberculosis transmission attributable to subclinical (asymptomatic) cases at the population level. The authors rely on existing survey data on household contacts of index cases with tuberculosis, and their respective symptomatic and bacteriological status. A solid, novel approach is proposed to incorporate this existing data to produce meaningful indicators of the relative importance of subclinical tuberculosis in sustaining the global tuberculosis epidemic. In addition to being of interest to tuberculosis epidemiologists, these results will be an important resource for policymakers in the ongoing debate about tuberculosis case finding and the role of symptom screening algorithms.

---

## [Decision Letter]

**Decision letter after peer review:**

Thank you for submitting your article "Estimating the contribution of subclinical tuberculosis disease to transmission: an individual patient data analysis from prevalence surveys" for consideration by *eLife*. Your article has been reviewed by 3 peer reviewers, and the evaluation has been overseen by a Reviewing Editor and Bavesh Kana as the Senior Editor. The reviewers have opted to remain anonymous.

Essential revisions:

1. The authors should run a sensitivity analysis on:

– The assumption that smear and symptom status are constant over time since this is almost certainly not the case. This could be accomplished, for example, by using the model described in sensitivity analysis #2 – by assigning estimates of per–unit–time relative infectiousness to each state in that model and then calibrating those estimates to the cumulative hazard ratios that are estimated in this study.

– The assumption that the background hazard of infection is the same across all households in a given survey/study. For example, do results change much if households with an index case have a 50% higher background hazard of infection?

– The durations of smear–positive and smear–negative TB are the same. For example, do results change much if smear–positive TB lasts twice as long as smear–negative TB?

2. Did the authors consider fitting a single cumulative hazard model for all 5 countries, that would allow only Λ_B (background cumulative hazard) to vary by country, rather than fitting 5 separate models and then pooling the estimates for the other lambdas after fitting? Does the former approach allow for estimation of the other λ's with more precision – unless the authors have reason to believe the λ's will vary by setting?

3. How were households with multiple TB cases treated in the analysis? If they were excluded, what effect is this likely to have on the analysis? This should be described somewhere (at least in the supplement) and the likely impact on results discussed.

4. Given how much the Philippines 1997 prevalence survey seems to be driving results (in terms of the larger sample size of household contacts), it would be useful to show how the percent of transmission results in Figure 3 look when applied to the Philippines 1997 survey data. Currently only one of the four surveys/studies used to derive the relative infectiousness estimates (Vietnam 2007) is used in the percent of transmission estimates.

Other issues that should be addressed:

1. The finding that subclinical TB is likely to be more transmissible than symptomatic TB is counterintuitive and warrants much more discussion.

2. It may be useful to clarify to readers how Mtb infection differs from subclinical TB (or TB more generally).

3. Figure 1 presents the odds ratios for infection of smear–positive vs. smear–negative (on the left) and clinical vs. subclinical TB (on the right). Figure 2 presents relative per–unit–time infectiousness estimates in the exact opposite way (smear–negative vs. smear–positive and subclinical vs. clinical, and in the opposite order). It would be clearer for readers if the presentation was more consistent across figures (and accompanying text).

4. Presenting figure 1 in terms of the relative cumulative hazard rather than (or in addition to) odds ratios would also help with the continuity between figures 1 and 2 (since the estimates of per–unit–time relative infectiousness used the relative cumulative hazards, not odds ratios).

5. Methods text should contain a bit more information on how the estimated durations of subclinical and clinical TB were used to convert the infection odds ratios into per–unit–time relative infectiousness estimates – this is clear in the appendix but not in the main text.

6. In the abstract, the term "individual population data" is a bit unclear.

7. Supplementary figure 2 may be mislabeled – for Bangladesh, it seemed that only smear–positive cases were included but the graph makes it look like only subclinical cases were included. Are the other panels similarly mislabeled?

8. p5 (line 7–9) Sentence could be made clearer. What does it mean there is a "disconnect"? Is this a pathological link? or causality? lack of evidence?

9. Results: The estimations of per unit time infectiousness of subclinical TB should be brought up front in the results, before the meta–analysis.

10. The discussion around the limitations of household contact and transmission assessment is very limited. This study inherits the limitations of the source studies, a more robust discussion around this issue would be beneficial.

11. Sensitivity of symptom screening algorithms is not discussed: TB disease, although split it into two categories is more likely a continuum.

12. Discussion page 15 (L11–L12): Is it relevant to mention SARS–CoV and Malaria here? Bringing asymptomatic transmission as a transversal subject across diseases without providing more epidemiological insights seems gratuitous. This should be removed.

13. The implications on policy and directly on recommendations around symptom screening in WHO plans are not commented on enough, this is where these results could be more useful.

14. page 16, line 32: The sentence "This is in keeping with increasing data showing that symptoms, in particular the classic TB symptom of cough, are not closely correlated to the amount of Mtb exhaled [28,29] " This is not completely consistent with Ref [28] by Theron et al. Nature Med 2020. The authors actually found that cough alone was a strong indicator for the measured transmissibility supplementary table 8. OR for CASS positivity for cough: 3.38 (1.90–5.99) despite the overall symptom score being negatively associated with transmissibility.

15. Supplementary Table 1: would be useful to add information on the definition of symptoms used to define subclinical against clinical disease.

16. Is there evidence from the literature regarding the subclinical transmission using results from whole genome sequencing? For example, Xu et al. (plos medicine) found in a study from Spain that 5/14 (35.7%) cases likely transmitted TB well before symptom onset. This can be discussed.

*Reviewer #1 (Recommendations for the authors):*

Slow declines in the global burden of TB are driven by a large gap between TB prevalence and the number of people with TB that are diagnosed and treated. Subclinical TB contributes to this "prevalence–notification gap"; a substantial (around 50%) of people with TB disease do not present with TB symptoms (or have fewer or milder symptoms that would not screen positive on a typical symptom screening questionnaire). The extent to which detecting and curing people with subclinical TB is important to reducing population–level TB transmission (and thus TB burden) is unclear, and depends both on infectiousness (how infectious people with subclinical TB are at any given point in time) and duration (how long they are expected to have TB before being detected or resolving on their own; and their associated future trajectory of infectiousness). Shedding light on this question could have a substantial impact on the global strategy for ending TB, including whether it is important for mass screening efforts (and other strategies to reduce the prevalence–notification gap) to detect those with subclinical TB and, if so, how to prioritize future diagnostics that best accomplish this (as the authors discuss in the last paragraph of the Discussion section).

In this study, the authors aimed to assess the infectiousness of people with subclinical TB (relative to symptomatic TB) at any given point in time; i.e., adjusting for the duration. To do this, they used individual– and household–level data from 4 cross–sectional surveys/studies that collected information on the prevalence of symptomatic TB, subclinical TB, and TB infection in households. The cumulative relative infectiousness of subclinical vs. symptomatic TB can be estimated by assessing how much higher the prevalence of TB infection is in households with a subclinical vs. symptomatic TB case. However, this estimate of cumulative relative infectiousness combines both pre–unit–time infectiousness and duration. To adjust for the duration and estimate per–unit–time relative infectiousness, the authors employed a cumulative hazard model that they then adjusted for estimates of the duration of symptomatic and subclinical TB; these duration estimates came from a mathematical model of TB natural history that is described in a different paper by Ku et al.

Strengths of the analysis include the separation of the duration of different forms of TB to estimate the per–unit–time relative infectiousness, as well as the Bayesian framework used to infer the cumulative relative hazards of infection from different forms of TB. Weaknesses include a number of assumptions made in the analysis that are unlikely to hold true and could bias results: the assumption that household index cases have continually had the same form of TB that they had when they were detected in a prevalence survey (or active case finding study), that smear–positive and smear–negative forms of TB are of the same duration, and that the background hazard of infection is the same for all households in a given survey/study. It would be useful for the authors to conduct a sensitivity analysis on these three assumptions. For example, in sensitivity analysis they estimate the durations of subclinical and symptomatic TB using a model that allows populations to transition between these states – could that model not be used to infer the per–unit–time relative infectiousness estimates? Additionally, historical data from the pre–antibiotic era (and a recent modeling study based on this data https://doi.org/10.1073/pnas.2211045119) suggests differential disease duration by smear status. In countries where a lot of TB is still diagnosed using smears (or was at the time of these prevalence surveys), differential detection/treatment probabilities could also lead to differential disease duration. This could have important implications for the analysis, given that symptom and smear status seem to be somewhat correlated in prevalence surveys (subclinical cases are much more likely to be smear–negative than smear–positive) and that smear status is more strongly predictive of infectiousness.

In all three cases, it is difficult to predict the direction and magnitude of potential bias on results, which is why sensitivity analysis is important and could strengthen the extent to which the authors' conclusions are justified by the analysis (currently this is somewhat unclear). Furthermore, the conclusion that subclinical TB is actually more transmissible than symptomatic TB is counterintuitive and warrants more discussion than is currently included in the study.

Despite these limitations, the broad conclusion that a substantial portion of TB transmission comes from subclinical cases will probably continue to hold even with adjustments to the analysis and additional sensitivity analyses. Even at the lower bound of the wide current estimated prediction interval, 27% of transmission coming from TB is substantial. This is an important finding, and thus additional analysis/revisions that can strengthen the robustness of this finding would be valuable.

Suggestions on analysis

– Suggest running sensitivity analysis on the assumption that smear and symptom status are constant over time since this is almost certainly not the case. This could be accomplished, for example, by using the model described in sensitivity analysis #2 – by assigning estimates of per–unit–time relative infectiousness to each state in that model and then calibrating those estimates to the cumulative hazard ratios that are estimated in this study.

– Suggest running sensitivity analysis on the assumption that the background hazard of infection is the same across all households in a given survey/study. For example, do results change much if households with an index case have a 50% higher background hazard of infection?

– Suggest running sensitivity analysis on the assumption that the durations of smear–positive and smear–negative TB are the same. For example, do results change much if smear–positive TB lasts twice as long as smear–negative TB?

– Did the authors consider fitting a single cumulative hazard model for all 5 countries, which would allow only Λ_B (background cumulative hazard) to vary by country, rather than fitting 5 separate models and then pooling the estimates for the other lambdas after fitting? I wonder if the former approach could allow for the estimation of the other lambdas with more precision – unless the authors have reason to believe the lambdas will vary by setting.

– How were households with multiple TB cases treated in the analysis? If they were excluded, what effect is this likely to have on the analysis? The effect could be to exclude those households that had the highest hazard of infection. This should be described somewhere (at least in the supplement) and the likely impact on results discussed.

– Given how much the Philippines 1997 prevalence survey seems to be driving results (in terms of the larger sample size of household contacts), it would be useful to show how the percent of transmission results in Figure 3 look when applied to the Philippines 1997 survey data. Currently only one of the four surveys/studies used to derive the relative infectiousness estimates (Vietnam 2007) is used in the percent of transmission estimates.

Suggestions on presentation

– The finding that subclinical TB is likely to be more transmissible than symptomatic TB is counterintuitive and warrants much more discussion (I don't believe this is currently discussed at all)

– Depending on the intended audience, it may be useful to clarify to readers how Mtb infection differs from subclinical TB (or TB more generally).

– Figure 1 presents the odds ratios for infection of smear–positive vs. smear–negative (on the left) and clinical vs. subclinical TB (on the right). Figure 2 presents relative per–unit–time infectiousness estimates in the exact opposite way (smear–negative vs. smear–positive and subclinical vs. clinical, and in the opposite order). It would be clearer for readers if the presentation was more consistent across figures (and accompanying text).

– Presenting figure 1 in terms of the relative cumulative hazard rather than (or in addition to) odds ratios would also help with the continuity between figures 1 and 2 (since the estimates of per–unit–time relative infectiousness used the relative cumulative hazards, not odds ratios).

– Methods text should contain a bit more information (like 1 sentence) on how the estimated durations of subclinical and clinical TB were used to convert the infection odds ratios into per–unit–time relative infectiousness estimates – this is clear in the appendix but not in the main text.

*Reviewer #2 (Recommendations for the authors):*

Emery et al., present for the first time quantitative estimates of the relative infectiousness of subclinical TB cases and the overall contribution of these to the global and country–specific (selected countries) ongoing TB transmission. The authors review and aggregate individual–level data from studies where household contacts and their index cases were assessed for their bacteriological and symptomatic status in order to reclassify and reanalyse household transmission accordingly. They produce aggregated estimates of the Odds Ratio of TB infection for households of subclinical TB index cases vs clinical index cases, and more importantly, they use a novel approach that uses cumulative hazard models to estimate the per unit infectiousness of subclinical TB. This approach accounts for important considerations like background TB transmission and duration of the infectious period. Their results (for ORs and infectiousness) show no significant advantage in the clinical index cases for infection. In light of these results, one can conclude that subclinical TB cases have no transmission disadvantage compared to symptomatic TB cases, and given its high prevalence in high–burden settings might be silently contributing disproportionately to the overall burden of TB. The implications of these results are that current approaches to active case finding and household investigation interventions put most of their efforts to identify symptomatic TB cases, potentially missing important sources of infection.

The main observable limitations of this work are "inherited" from the sources of data, namely:

1. Household contacts of index cases are a very indirect (although useful) way of assessing TB transmission and infectiousness: The direction of transmission (index case to contact) can never truly be asserted, not even can it be ruled out to have occurred from external sources. The authors only briefly mention this but it is the main source of uncertainty when discussing TB household transmission investigation. The approach here proposed does not address these points, and thus it remains a gap and a challenge in the research field.

2. Assessment of TB symptoms is in itself a type of diagnostic tool with a measurable level of performance (sensitivity/specificity). This can drastically alter the estimation of relative infectiousness by clinical/subclinical status. The authors don't mention this and it is not immediately clear how this was done in the source studies.

I have enjoyed having the opportunity of reviewing this manuscript, which I think can be a widely used resource for policymakers, mathematical modellers, and epidemiologists working on TB. The main advantages and solid points I found are summarised in my public review and assessment. Here I add some points that I think need to be addressed to improve the presentation and conveyance of the message.

– p5 (line 7–9) Sentence could be made clearer. What does it mean there is a "disconnect"? Is this a pathological link? or causality? lack of evidence?

– Results: In my view, the estimations of per unit time infectiousness of subclinical TB should be brought up front in the results, before the meta–analysis.

– The discussion around the limitations of household contact and transmission assessment is very limited. As I said before, this study inherits the limitations of the source studies but a more robust discussion around this issue would feel healthier (Kendall editorial note, etc..)

– Sensitivity of symptom screening algorithms is not discussed: TB disease, although we split it into two categories is more likely a continuum, and how good we get at identifying those signs determines how much we will trust these results in the future.

– Discussion page 15 (L11–L12): I wonder how relevant is to mention SARS–CoV and Malaria here. Bringing asymptomatic transmission as a transversal subject across diseases without providing more epidemiological insights seems gratuitous. I would suggest either removing or expanding.

– The implications on policy and directly on recommendations around symptom screening in WHO plans are not commented on enough, and I believe this is where these results could be more useful.

*Reviewer #3 (Recommendations for the authors):*

The investigators applied a statistical framework to synthesize information from prevalence surveys of active TB disease (regardless of symptoms) and latent tuberculosis infection among household contacts of index cases from tuberculosis surveys. They estimated an odds ratio of 1.2 (95% CI: 0.6–2.3) for infection in a household with a clinical versus subclinical index case (not considering the duration of the disease). They estimated that 68% (27–92%) of global tuberculosis transmission was from subclinical TB.

The major strength of the study is the ability to collect relevant information from prevalence surveys and integrate it using a sophisticated statistical framework. The result is novel in the sense that no such estimation has been provided before.

One major limitation of the study, as acknowledged by the authors, is the very wide confidence interval (27–92%) for the estimated global proportion of transmission contributed by subclinical disease. The wide 95% CI was consistent with a range from modest to very large impact of subclinical TB and therefore has different policy implications for tuberculosis control. It will be useful if the authors could indicate the major source of uncertainty behind this confidence interval. In other words, where and how should we collect more data to improve the precision of this estimation procedure?

Another limitation of the study is the lack of consideration of the conversion between subclinical and clinical TB status (i.e., not a one–way street) in the cumulative hazard model. Its impact on the main finding (the transmission contribution from subclinical disease) remains unclear. Considering the natural history of TB is usually from subclinical to clinical (with conversion and reversion along the way), it is anticipated that the cumulative latent infection among household contacts of a clinical TB index case would likely be contributed by both the clinical period as well as the subclinical period. This "misclassification" of clinical vs. subclinical TB seems to bias the relative infectiousness (clinical vs. subclinical) towards the null value of OR=1, under the hypothesis that clinical TB is more infectious than subclinical TB. In other words, it might therefore underestimate the relative transmissibility of clinical TB.

One of the major results of the study was that an estimated 68% (27–92%, 95% PrI) of global transmission was from subclinical TB. Given that there is substantial between–country variation in the % of subclinical TB among prevalent cases (Figure 3A) and variation in the duration of subclinical TB (see. Ku C–C et al. 2021, citation #32 of this paper), the country–level heterogeneity of the % transmission from subclinical cases should be acknowledged when making inference on the global estimate, accounting for the burden of disease in each geography. The current global estimate was based on the pooled parameter values from 14 countries (not including major countries like India and China) without formal consideration of the burden of disease. I, therefore, found the claim of global estimate a bit too strong.

Despite all these limitations, the present research is considered novel and will contribute to the field of TB epidemiology. It will generate more discussion (and debate) on the importance of detecting and treating subclinical TB. Better data and further refinement of the analytic framework will generate more precise information which will prove to be useful for the revision of TB control strategies.

---

## [Author Response]

Essential revisions:1. The authors should run a sensitivity analysis on:– The assumption that smear and symptom status are constant over time since this is almost certainly not the case. This could be accomplished, for example, by using the model described in sensitivity analysis #2 – by assigning estimates of per–unit–time relative infectiousness to each state in that model and then calibrating those estimates to the cumulative hazard ratios that are estimated in this study.

The cumulative hazard approach we have used in this work was chosen to suit the available data and corresponding study designs. We agree that the assumption of a single disease phase (e.g. just subclinical, smear–negative) is a limitation of this approach and acknowledge this in our discussion (see Page 18, Line 11).

We appreciate the suggestion of changing the approach to a more dynamic model. This would ultimately change the methodology of the paper however and, in the absence of additional data with which to fit the more complex model, would rely on additional assumptions. While an interesting approach, we prefer to retain the current balance between data, assumptions and model complexity.

See Page 18, Line 11 for an expanded discussion of this limitation:

“An important limitation of our cumulative hazard model is the assumption that index cases only ever had the disease type they were diagnosed with during screening (e.g. sputum smear–positive, subclinical). Instead, it is more likely that individuals will fluctuate between being, for example, subclinical and clinical [31]. The impact such additional dynamics would have on our results remains uncertain, since they would depend on the detailed model of tuberculosis natural history assumed. Such a model would require additional data to prevent the need for additional assumptions.”

– The assumption that the background hazard of infection is the same across all households in a given survey/study. For example, do results change much if households with an index case have a 50% higher background hazard of infection?

Thank you for the suggestion, we have included this as ‘Sensitivity analysis 3’.

See Page 10, Line 29 for a description of the methods:

“Sensitivity analysis 3: To explore the impact of a differential background risk of infection amongst households with and without index cases, the analysis was repeated assuming a 50% increase in the background risk of infection for those households with an index case.”

See Page 15, Line 31 for a description of the results:

Sensitivity analysis 3: The above analysis was repeated assuming that households with an index case have a 50% greater background risk of infection than households with no index case. Affected results are shown in Supplementary Figure 8. The infectiousness of subclinical TB per unit time relative to clinical TB increased to 2.44 (0.60–10.06, 95% PrI), with corresponding values of 74% (29–95%, 95% PrI) of global transmission from subclinical TB, ranging from 53% (20–85%, 95% PrI) in Nigeria to 88% (61–97%, 95% PrI) in Mongolia.

See Page 17 in the Supplementary Materials for plots of the results.

– The durations of smear–positive and smear–negative TB are the same. For example, do results change much if smear–positive TB lasts twice as long as smear–negative TB?

We have also run this sensitivity analysis as ‘Sensitivity analysis 4’.

Please see Page 11, Line 1 for a description of the methods:

“Sensitivity analysis 4: Instead of assuming equal durations for sputum smear–positive and smear–negative TB, the above analysis was repeated assuming that sputum smear–positive TB has twice the duration of smear–negative TB.”

And page 16, Line 6 for a description of the results:

“Sensitivity analysis 4: The above analysis was repeated assuming sputum smear–positive TB has twice the duration of smear–negative TB. Affected results are shown in Supplementary Figure 9. The infectiousness of subclinical TB per unit time relative to clinical TB was largely unchanged at 1.94 (0.63–6.16, 95% PrI), with corresponding values of 70% (29–93%, 95% PrI) of global transmission from subclinical TB, ranging from 49% (21–79%, 95% PrI) in Nigeria to 86% (63–96%, 95% PrI) in Mongolia.”

See Page 18 in the Supplementary Materials for plots of the results.

2. Did the authors consider fitting a single cumulative hazard model for all 5 countries, that would allow only Λ_B (background cumulative hazard) to vary by country, rather than fitting 5 separate models and then pooling the estimates for the other lambdas after fitting? Does the former approach allow for estimation of the other λ's with more precision – unless the authors have reason to believe the λ's will vary by setting?

An interesting suggestion, which we have included as ‘Sensitivity analysis 5’.

Methods are described on Page 11, Line 5:

“Sensitivity analysis 5: In the main analysis each study was modelled separately, with the results combined using meta–analyses. As a sensitivity we model all studies simultaneously, assuming local background risks of infection for each study and global values across all studies for the remaining cumulative hazards.”

And see Page 16, Line 14 for a description of the results:

“Sensitivity analysis 5: The above analysis was repeated with all studies modelled simultaneously, assuming local background risks of infection for each study and global values across all studies for the remaining cumulative hazards. Affected results are shown in Supplementary Figure 10. The infectiousness of subclinical TB per unit time relative to clinical TB decreased to 1.39 (0.50–4.02, 95% PrI), with corresponding values of 58% (20–88%, 95% PrI) of global transmission from subclinical TB, ranging from 34% (15–62%, 95% PrI) in Nigeria to 76% (52–91%, 95% PrI) in Mongolia.”

See Page 19 in the Supplementary Materials for plots of the results.

We have also added some comments on all of the sensitivities in the discussion (see Page 16, Line 33):

“Our results were relatively robust to the sensitivities that were performed. In two cases, that is the removal of two studies (sensitivity analysis 1) and the use of a single model to account for all studies (sensitivity analysis 5), our estimates for the relative infectiousness of subclinical TB relative to clinical TB and the proportion of transmission from subclinical TB were lower than in the primary analysis. Our qualitative results and conclusions remain unchanged however. All other sensitivities resulted in higher estimates.”

3. How were households with multiple TB cases treated in the analysis? If they were excluded, what effect is this likely to have on the analysis? This should be described somewhere (at least in the supplement) and the likely impact on results discussed.

Households with multiple index cases were excluded from the analysis, as the premise of the household contact approach is to ascribe the incremental infections to the household index case. This assumption becomes difficult to interpret if there are multiple household index cases (e.g. in terms of unknown differences in disease onset), resulting in an inability to ‘divide’ the incremental infections in the household between the index cases. Excluding these households results in a slight loss of power (co–prevalent household index cases were below 10% across studies), but we feel this is worth the resulting clarity of the association.

We have added the following in the text (Page 3, Line 4 of the Supplementary Materials):

“Households with multiple co–prevalent index cases were excluded from the analysis, to retain the premise of the analysis which links infections above the community level to the characteristics of the single index case. Co–prevalent cases were absent in one study (ACT3 [5]) and below 10% in other studies, limiting the impact on power or introduction of bias.”

4. Given how much the Philippines 1997 prevalence survey seems to be driving results (in terms of the larger sample size of household contacts), it would be useful to show how the percent of transmission results in Figure 3 look when applied to the Philippines 1997 survey data. Currently only one of the four surveys/studies used to derive the relative infectiousness estimates (Vietnam 2007) is used in the percent of transmission estimates.

Thank you for the suggestion. Whilst we agree that this would be an interesting addition, given the number of surveys used and their range of proportions subclinical and sputum smear–positive (see Figure 3), the impact on the proportion of global transmission from subclinical TB is likely to be marginal, leaving our qualitative results and conclusions unchanged. We have instead focused on the three additional sensitivity analyses detailed above.

Other issues that should be addressed:1. The finding that subclinical TB is likely to be more transmissible than symptomatic TB is counterintuitive and warrants much more discussion.

We agree that subclinical TB is unlikely to be more transmissible than clinical TB and have added further text to the discussion to address this point.

See Page 17, Line 21:

“Indeed, the paucity of the data provides an estimate that is consistent with subclinical TB being more infectious than clinical TB. Whilst we consider this to be implausible, we have avoided introducing priors that rule out this possibility. Instead we would emphasise that our results reflect the uncertainty of the data. The lower bound of our estimate precludes subclinical TB being significantly less infectious than clinical TB, while there is no evidence against subclinical TB being as infectious as clinical TB.”

2. It may be useful to clarify to readers how Mtb infection differs from subclinical TB (or TB more generally).

Thank you. We have added a line in the introduction to this effect.

See Page 4, Line: 31:

“We distinguish both disease states from Mtb infection, whereby individuals may test positive on a Tuberculin Skin Test (TST) or Interferon–Γ Release Assay (IGRA) but do not have bacteriologically–confirmed disease.”

3. Figure 1 presents the odds ratios for infection of smear–positive vs. smear–negative (on the left) and clinical vs. subclinical TB (on the right). Figure 2 presents relative per–unit–time infectiousness estimates in the exact opposite way (smear–negative vs. smear–positive and subclinical vs. clinical, and in the opposite order). It would be clearer for readers if the presentation was more consistent across figures (and accompanying text).

Thank you for your comment. We agree this was inconsistent. Figure 1 has been altered accordingly to show subclinical vs. clinical on the left and sputum smear–negative vs. smear–positive on the right (please see Page 12).

4. Presenting figure 1 in terms of the relative cumulative hazard rather than (or in addition to) odds ratios would also help with the continuity between figures 1 and 2 (since the estimates of per–unit–time relative infectiousness used the relative cumulative hazards, not odds ratios).

Whilst we agree that presenting cumulative hazards would be closer to the modelling framework, we chose odds ratios to complement the modelling results, since they are more intuitive for a wide range of potential readers. We feel this helps to broaden the readership and understanding of the work and would therefore prefer to keep this metric.

5. Methods text should contain a bit more information on how the estimated durations of subclinical and clinical TB were used to convert the infection odds ratios into per–unit–time relative infectiousness estimates – this is clear in the appendix but not in the main text.

Further details have been added to the methods.

See Page 8, Line 10:

“To infer the infectiousness of subclinical TB per unit time relative to clinical TB from our posterior estimate for the subclinical relative cumulative hazard r_s_ we note that, assuming constant hazards, the relative cumulative hazards from index cases will depend on the product of the relative per unit time infectiousness and relative durations of infectiousness. We assume that per unit time infectiousness depends on symptom status and sputum smear–status, whilst durations of infectiousness depends on symptom status only. It follows then that: rs=αsγs,r_=α_, where *α*_s_ and *α*_–_ are the per unit time infectiousness of subclinical relative to clinical index cases and sputum smear–negative relative to smear positive index cases, respectively, and γ_s_ is the duration of infectiousness for subclinical relative to clinical index cases.

Finally, we sampled from the posterior estimate for the subclinical relative cumulative hazard and an assumed duration of disease for subclinical index cases relative to clinical index cases, providing a median and 95% equal tailed posterior estimate for the relative infectiousness of subclinical index cases relative to clinical index cases for each study separately. Finally, we provide a summary estimate by mixed–effects meta–analysing the individual estimates across the separate studies. Analogous results are presented for the relative infectiousness per unit time of sputum smear–negative TB relative to smear–positive TB.”

6. In the abstract, the term "individual population data" is a bit unclear.

This has been edited.

See Page 2, Line 28:

“We collated individual–level data on representative populations for analysis…”

7. Supplementary figure 2 may be mislabeled – for Bangladesh, it seemed that only smear–positive cases were included but the graph makes it look like only subclinical cases were included. Are the other panels similarly mislabeled?

Thank you for the comment – this was indeed mislabelled and has been edited. All other panels are correct.

See Page 11 of the Supplementary Materials.

8. p5 (line 7–9) Sentence could be made clearer. What does it mean there is a "disconnect"? Is this a pathological link? or causality? lack of evidence?

This has been edited.

See Page 5, Line 9:

“Indeed, whilst recent empirical studies have suggested that tidal breathing may contribute significantly to Mtb transmission [27], exhalation of infectious aerosols appears unrelated to the presence of symptoms [28] or cough frequency [29] in TB patients.”

9. Results: The estimations of per unit time infectiousness of subclinical TB should be brought up front in the results, before the meta–analysis.

We agree that the results for the per unit time infectiousness of subclinical TB could be made more prominent. However, moving Figure 2 (estimates of per unit time infectiousness) above Figure 1 (odds ratios of the underlying data) would be inconsistent with the order of the methods. We instead have edited the text by introducing subsections ‘Data’ and ‘Estimating the relative infectiousness of subclinical TB’ to make the results more easily identifiable whilst preserving the order of the methods.

See Page 11, Line 12 for the first new subsection:

“Data”.

See Page 12, Line 10 for the second new subsection:

‘Estimating the relative infectiousness of subclinical TB’.

10. The discussion around the limitations of household contact and transmission assessment is very limited. This study inherits the limitations of the source studies, a more robust discussion around this issue would be beneficial.

We have added additional text to the discussion of this limitation to acknowledge that our work necessarily inherits all such limitations. Given other additions to the discussion as a result of peer review, we feel any further comment would make the Discussion section prohibitively long.

See Page 18, Line 4:

“Such household contact studies are therefore liable to biases and our study necessarily inherits such limitations. For example…”

11. Sensitivity of symptom screening algorithms is not discussed: TB disease, although split it into two categories is more likely a continuum.

A paragraph has been added to the discussion to this effect.

See Page 18, Line 26:

“We defined subclinical and clinical TB as being culture and/or NAAT positive and responding negatively or positively to an initial symptom screen, respectively. In practice subclinical and clinical TB are part of a continuous spectrum and alternative definitions could be defined according to different criteria. Here we have used the definition most closely aligned with the methodology of the majority of prevalence surveys, which is consistent with other studies of subclinical TB [18] and pragmatic for inclusion of future surveys.”

12. Discussion page 15 (L11–L12): Is it relevant to mention SARS–CoV and Malaria here? Bringing asymptomatic transmission as a transversal subject across diseases without providing more epidemiological insights seems gratuitous. This should be removed.

We agree that whilst evidence for asymptomatic transmission in SARS–CoV–2 and Malaria does not provide evidence for such transmission in TB, we mention them in the discussion to make the broader point that asymptomatic transmission would not be unique to TB. We have edited the text to clarify this point.

See Page 17, Line 17:

“More broadly, recent work on SARS–CoV–2 and malaria have similarly shown how ‘asymptomatic’ or ‘subpatent’ infections can be important drivers of transmission [45–47], meaning a role for asymptomatic transmission would not be unique to TB.”

13. The implications on policy and directly on recommendations around symptom screening in WHO plans are not commented on enough, this is where these results could be more useful.

We have added a reference to the WHO screening guidelines [1] in the discussion (see Page 19, Line 31):

“As our results show that subclinical TB likely contributes substantially to transmission, an increased emphasis on symptom agnostic screening in, for example, the TB screening guidelines [58] should be considered, as should the inclusion of subclinical TB in the planned update of WHO case definitions. Target Product Profiles for diagnostic tools should consider all infectious TB, regardless of whether individuals are experiencing or aware of symptoms, and interventions using such tools should be critically evaluated for their impact on Mtb transmission and cost–effectiveness.”

14. page 16, line 32: The sentence "This is in keeping with increasing data showing that symptoms, in particular the classic TB symptom of cough, are not closely correlated to the amount of Mtb exhaled [28,29] " This is not completely consistent with Ref [28] by Theron et al. Nature Med 2020. The authors actually found that cough alone was a strong indicator for the measured transmissibility supplementary table 8. OR for CASS positivity for cough: 3.38 (1.90–5.99) despite the overall symptom score being negatively associated with transmissibility.

We have removed the reference to Theron et al. Nature Med 2020 (Ref [28] in the above).

See Page 19, Line 19:

“This is in keeping with increasing data showing that symptoms, in particular the classic TB symptom of cough, are not closely correlated to the amount of Mtb exhaled [29]…”

15. Supplementary Table 1: would be useful to add information on the definition of symptoms used to define subclinical against clinical disease.

These have been added to Supplementary Table 1.

See Page 4 in the Supplementary Materials.

16. Is there evidence from the literature regarding the subclinical transmission using results from whole genome sequencing? For example, Xu et al. (plos medicine) found in a study from Spain that 5/14 (35.7%) cases likely transmitted TB well before symptom onset. This can be discussed.

We have added this study [2] to the discussion.

See Page 17, Line 13:

“Our results are also in keeping with recent results from whole genome sequencing [45], in which 36% of individuals likely transmitted Mtb before symptom onset, assuming a linear SNP mutation rate.”

We have also added another study to the discussion that was published whilst this paper was under review [3].

See Page 17, Line 7:

“Using data from the 2007 Viet Nam prevalence survey however, [43] find that among children aged 6–10 years, those living with clinical, smear–positive tuberculosis patients, and those living with subclinical, smear–positive tuberculosis patients, had similarly increased risks of TST positivity compared with those living without tuberculosis patients.”

References

[1] World Health Organization. WHO consolidated guidelines on tuberculosis – Module 2: Systematic screening for tuberculosis disease. Genève: World Health Organisation 2021. https://apps.who.int/iris/handle/10665/341426 (accessed 9 Sep 2021).

[2] Xu Y, Cancino–Muñoz I, Torres–Puente M, et al. High–resolution mapping of tuberculosis transmission: Whole genome sequencing and phylogenetic modelling of a cohort from Valencia Region, Spain. PLoS Med 2019;16:e1002961. doi:10.1371/journal.pmed.1002961

[3] Nguyen HV, Tiemersma E, Nguyen NV, et al. Disease Transmission by Patients With Subclinical Tuberculosis. Clin Infect Dis 2023;76:2000–6. doi:10.1093/cid/ciad027